# Visual attention to emotional faces in adolescents with social anxiety disorder receiving cognitive behavioral therapy

Jens Högström[1]*, Martina Nordh[1], Miriam Larson Lindal[2], Ebba Taylor[2], Eva Serlachius[1], Johan Lundin Kleberg[1,3]

**1** Centre for Psychiatry Research, Department of Clinical Neuroscience, Karolinska Institutet, & Stockholm Health Care Services, Stockholm County Council, CAP Research Center, Gävlegatan, Stockholm, Sweden, **2** Division of Psychology, Department of Clinical Neuroscience, Karolinska Institutet, Stockholm, Sweden, **3** Uppsala Child and Baby Lab, Department of Psychology, Uppsala University, Uppsala, Sweden

* jens.hogstrom@ki.se

**Data Availability Statement:** Data cannot be shared publicly because of restrictions posed by the regional ethical review board in Stockholm. Data are however available from the corresponding

## Abstract

Social anxiety disorder (SAD) is a psychiatric condition that often onsets in childhood. Cognitive models underline the role of attention in the maintenance of SAD, but studies on youth populations are few, particularly those using eye tracking to measure attention. Cognitive behavioral therapy (CBT) for SAD includes interventions targeting attention, like exposure to eye contact, but the link between CBT and attention bias is largely unexplored. This study investigated attention bias in youth with SAD and the association with outcome from CBT. Latency to attend to pictures of faces with different emotions (vigilance) and latency to disengage from social stimuli (avoidance) was examined in $N = 25$ adolescents (aged 13–17) with SAD in relation to treatment outcome. Vigilance was operationalized as the time it took to relocate the gaze from a central position to a peripherally appearing social stimulus. The latency to disengage from a centrally located social stimulus, when a non-social stimulus appeared in the periphery, was used as a proxy for avoidance. Attention characteristics in the SAD group were compared to non-anxious (NA) controls ($N = 22$). Visual attention was measured using eye tracking. Participants in both the SAD and NA groups were vigilant towards angry faces, compared to neutral and happy faces. Similarly, both groups disengaged attention faster from angry faces. Adolescents with SAD who disengaged faster from social stimuli had less social anxiety after CBT. The results indicate that anxious youth display a vigilant-avoidant attention pattern to threat. However, partly inconsistent with previous research, the same pattern was observed in the NA group.

## Introduction

Social anxiety disorder (SAD) is one of the most common anxiety disorders with a lifetime prevalence of approximately 12% [1]. Proposed etiological and maintaining factors of social anxiety include different cognitive biases, such as tendencies to interpret social situations negatively, excessive self-focused scrutiny and a propensity to make inaccurate inferences from

author (jens.hogstrom@ki.se) and from the Clinical Research Unit within the Child and Adolescent Psychiatry in Stockholm (bupkfe.slso@sll.se), for researchers who meet the criteria for access to confidential data.

**Funding:** This work was supported by the Stockholm County Council and the regional agreement on medical training and clinical research between Stockholm County Council and the Karolinska Institute (forskningsstod.vmi.se) under Grant 20150032 and The Swedish Research Council for Health, Working Life and Welfare (www.forte.se/en/) under Grant 2014-4052. Author ES received both grants. The funders had no role in study design, data collection and analysis, decision to publish, or preparation of the manuscript.

**Competing interests:** The authors have declared that no competing interests exist.

internal information (such as anxiety) about how one is perceived by others [2]. Cognitive and behavioral models also emphasize the importance of attention processes in the development and maintenance of SAD [3–5], but there is no clear consensus on the specific nature of these attention characteristics. For instance, the model by Rapee and Heimberg [4] proposes that individuals with SAD are more vigilant toward threat whereas the Clark and Wells model [3] emphasizes that individuals with SAD are more prone to avoid threat-related information. Both models, however, agree that attention biases along with atypical processing of information from our surroundings contribute to the maintenance of social anxiety. Findings from experimental as well as non-experimental studies point to the importance of these factors in social anxiety. Self-report studies assessing focus of attention have, for instance, shown that there is a link between self-focused attention and SAD but the validity of assessing direction of attention with questionnaires in youth populations has been questioned [2]. Therefore, the majority of studies in the attention field have used experimental designs to capture how individuals with SAD observe the outside world. A better understanding of how socially anxious individuals perceive social situations could help explain how negative impressions of social environments arise. These insights could potentially provide hypotheses about how current evidence-based treatments, such as cognitive behavioral therapy (CBT), could be improved.

The majority of studies on attention biases in anxiety disorders have been conducted with the dot-probe paradigm. In this task, participants are required to detect a visual cue and indicate with a manual response whether it occurs at a location previously occupied by a threatening stimulus (e.g., an angry face) or a non-threatening stimulus. Relatively quicker responses to locations previously occupied by a threatening stimulus are taken as evidence for vigilance, whereas relatively slower responses to these stimuli are taken as evidence for avoidance [6]. Several authors have noted problems with this task (e.g., [7]). First, attention has to be dichotomously classified as *either* vigilant or avoidant, although research indicates that both processes are dynamic, and can occur within the same trial [8]. Second, in the most common applications, the dot-probe task cannot disentangle *biased attention toward* threatening stimuli from *difficulties with disengaging* from threat [7]. This is problematic since the brain functions underlying these two attention processes are distinct and dissociable (e.g., [9]). Third, individual variation in manual response accuracy and speed is a potential confounder. Gaze-based methods such as eye tracking have therefore become more frequently used, providing new insights in this field of research. One of the advantages with eye tracking is that it enables uninterrupted assessment of visual attention in the presence of emotionally valenced information [10].

Experimental research on attention patterns in anxious individuals has mainly focused on trying to find vigilant and avoidant biases. This research has shown that across different anxiety disorders and across different methods for measuring attention, there is an overall tendency to attend quicker to threat, in anxious adults [11, 12] as well as in children [13]. However, for children, this vigilance to threat seems to be present in non-anxious individuals as well, and the effect has been shown to be moderated by age across studies [13]. This meta-analysis showed that it was more likely to find attention bias differences between anxious and non-anxious groups in older youth samples, than in samples with younger children. The reason for this is thought to be related to a general tendency for younger children to respond to information from the surrounding in a bottom-up and valence-driven way. With maturation though, typically developed children learn to inhibit automatic responding and gradually become less reactive to threatening stimuli, whereas those with an anxiety disorder are more likely to maintain the biased attention towards threat [13].

In eye-tracking studies, vigilance (biased attention toward threat) is often defined as a proneness to orient faster, or more often, to threatening stimuli, as opposed to non-

threatening stimuli. Avoidance (biased attention away from threat) is conversely commonly defined as an inclination to saccade faster, or more frequently, to non-threatening stimuli compared to threatening stimuli [8]. Another process of presumed importance for maintenance of anxiety is *attentional disengagement*. In eye-tracking studies, disengagement can be conceptualized as the latency to detach the gaze from a stimulus (e.g., [14]]. A relatively longer latency to disengage from a fixated stimulus indicates impaired disengagement whereas shorter latency infers avoidance of, for instance, a threatening stimulus. Difficulty with disengagement has been shown to contribute to social anxiety in adults and the ability to disengage adequately from threats is thought to prevent an individual from ruminating too much on negative aspects of the surrounding [15, 16]. Attentional avoidance of social threat cues are also believed to exacerbate anxiety in youth, but through a tendency to miss out on important social information and thereby risking an exaggeration of the perceived threat [5].

There are few youth studies that have focused on attentional biases such as vigilance and avoidance in SAD specifically, as the common procedure in the field has been to include participants with mixed anxiety disorders. To use one form of presumably threatening stimuli for individuals with different anxiety disorders could be problematic as disorder-congruent threats are known to be associated with more attention bias than generally threatening stimuli [17]. One of the few diagnosis-specific studies using eye tracking to compare vigilance and avoidance in a homogenous sample of children with SAD, with a group of non-anxious controls, found vigilance towards angry faces in both groups, although children with SAD displayed a stronger bias when subjected to induced anxiety [18]. Another study on children with SAD, using the dot-probe task, showed that those with more severe social anxiety were vigilant towards angry faces, relative to neutral faces, whereas those with milder social anxiety tended to avoid threatening stimuli. Children in the non-anxious control group did not show any attention bias in either direction, i.e., they were as likely to direct their gaze toward angry as well as to neutral faces when they were presented in pairs [19]. A third study compared children with SAD to a healthy control group, using eye tracking, and found that both groups tended to avoid angry faces, relative to neutral faces [20].

Cognitive behavioral models of SAD as well as the experimental research aiming to verify these models have predominantly focused on trying to find evidence for *either* vigilance or avoidant attention biases, but a recent hypothesis suggests that anxious individuals might display *both* biases. First, an initial vigilance to threat, and then a later, top-down regulated, avoidance bias (the vigilance-avoidance hypothesis; [21, 22]). This multistage process would likely serve to first detect threat and then to avoid interaction with the perceived danger [23]. One study using eye tracking found support for the vigilance-avoidance hypothesis in young adults highly fearful of negative evaluation, a core feature in SAD [24], and another eye-tracking study on youth with mixed anxiety disorders (including SAD) did also show a vigilant-avoidant pattern of attention during a 3000 ms presentation of emotional stimuli, although this pattern was also observed in the non-anxious control group [25]. A review by Chen and Clarke [8] on gaze-based attention in SAD concluded that both vigilance and avoidance seems to be occurring during processing of emotional social stimuli, although there is no evident time-course across studies as vigilance and avoidance have been found at both initial (0–1000 ms) and later (>1000 ms) stages of processing. This could indicate that results from visual attention studies on SAD might differ depending on: 1) the age of the target population, 2) whether or not anxiety is induced during the experiment, 3) differences in eye-tracking paradigms (e.g., competing stimuli face pairs and gap-overlap paradigms), 4) the type of presented emotional stimuli, and, 5) on stimulus presentation durations. To draw firm conclusions about the specific nature of visual attention bias in youth SAD, more studies are needed to clarify the importance of these potentially moderating variables and methodological aspects. It has

furthermore been suggested that eye tracking and assessment of visual attention in SAD should be incorporated into clinical trials evaluating the efficacy of psychosocial interventions for SAD, in order to identify biases that may be predictive of better or worse outcomes [8]. Such research could help delineate the functional role of attention biases in the maintenance of SAD.

CBT has been shown to be an effective treatment for youth with SAD [26] and to meet the demand for evidence-based treatments, emerging research has focused on Internet-delivered CBT (ICBT; [27]). Recently, ICBT has showed promising effects in the treatment of youth with SAD [28, 29]. Several attempts have been made to link attention bias to the outcome of CBT showing that vigilance towards threat in anxious children predicts greater symptom reductions [30, 31]. The same has been reported in a homogenous sample of adults with SAD [32], and a suggested explanation is that those who attend more to threat during exposure exercises will experience more habituation and thereby greater alleviation of symptoms. The opposite result was, however, found by Legerstee and colleagues [33], who reported that children with mixed anxiety disorders (including SAD), with a pre-treatment bias towards severe threat, were less likely to respond well to CBT. All abovementioned studies used reaction-time measures but an association between vigilance to threat and poor treatment outcome was also shown in an eye-tracking study where adults with SAD received CBT [34]. In summary, it is not clear if an attention bias *towards* or *away* from threat, prior to entering therapy for SAD, will increase the likelihood of a positive treatment outcome.

Another sparsely studied topic is if psychosocial treatments such as CBT can reduce attention biases. CBT includes components that target different aspects of attention, e.g., redirection of attention toward threat (during exposure) and cognitive restructuring that aim to change the appraisal of threats in the environment. Studying if attention is modifiable by treatment is therefore important as it may inform us about whether attention bias in SAD is a stable trait or if it could be targeted in treatment as a means to reduce anxiety symptoms [35]. If the former is correct, a better understanding of attention processes in SAD may inform etiological models. If the latter is correct, targeting social attention may be an important objective for future psychosocial treatments of SAD. Research on adults with SAD has shown that selective attention to threat-related information decreases after CBT [36, 37]. In a clinical trial where children with mixed anxiety disorders were treated with CBT, it was shown that selective attention towards threatening stimuli was reduced during treatment, but only for those who responded to treatment [33]. Waters and colleagues [30], on the other hand, found that anxious children with a pre-treatment vigilance to threat remained vigilant after CBT whereas those with an initial attention bias away from threat experienced a significant reduction of their bias over the course of treatment. No prior study, however, has studied the effect of CBT on attention bias in a homogenous group of youth with SAD.

In the present study, vigilance toward social stimuli and latency to disengage (avoidance) from social stimuli were measured in adolescents with SAD as well as in a group of non-anxious controls, using eye tracking. The social stimuli consisted of pictures of angry, neutral as well as happy faces, as previous studies have indicated that not only threatening but also positive emotional stimuli can be negatively perceived in SAD [8]. The SAD group then received ICBT and visual attention was measured again at post-treatment and at a 6-month follow-up. We aimed to investigate, 1) if adolescents with SAD are more vigilant towards socially threatening stimuli compared to non-anxious controls, and 2) if adolescents with SAD are quicker to disengage (i.e., are more avoidant) when presented with socially threatening stimuli, compared to non-anxious controls. The study also aimed to examine if 3) vigilance or disengagement latency predicts the outcome of ICBT for adolescents with SAD, and 4) if vigilance towards, or disengagement/avoidance from, social stimuli change over the course of ICBT.

## Method

### Participants

Adolescents in the SAD group ($N$ = 25, age 13–17) were initially recruited within the frames of a clinical trial evaluating the efficacy of ICBT (see [28]). All adolescents included in the clinical trial ($N$ = 30) were asked if they also wanted to participate in a study about visual attention and social anxiety. Twenty-seven individuals with SAD agreed to participate initially but due to a technical failure, eye-tracking data was lost for two participants and hence 25 participants formed the SAD group. All participants fulfilled DSM-5 criteria for a principal SAD diagnosis according to the Mini International Neuropsychiatric Interview for Children and Adolescents (MINI-KID; [38]). In the SAD group, 14 participants (56%) had a comorbid disorder and the most common ones were specific phobia (7 participants, 28%) and generalized anxiety disorder (5 participants, 20%). Participants in the non-anxious (NA) group ($N$ = 22, age 13–18) were recruited from 450 randomly selected adolescents in the population register. This register is administered by the Swedish Tax Agency and provides services to government institutions, including universities, that are in need of population data such as personal identification numbers and addresses to groups of individuals. Participants in the NA group were matched with the SAD participants on age and gender. Participation in the NA group was eligible if an individual did not meet criteria for any psychiatric disorder after being screened with the same diagnostic interview used in the SAD group (MINI-KID; [38]). The sample size was similar to, or slightly larger than, most previous eye-tracking studies investigating social attention [11], with 80% power, given alpha 0.05, two-tailed, to detect medium effect sizes (corresponding to Cohen's $d$ = .7 for comparisons between the SAD and NA group and $d$ = .6 for within-group analyses in the SAD group).

As expected, individuals in the NA group had significantly less social anxiety, measured with SPAI-C, compared to the SAD group but there were no other significant differences between the groups (see Table 1). The study was approved by the regional ethical review board in Stockholm.

### Procedure

For the SAD group, the clinical assessment and eye-tracking experiment were conducted prior to treatment, at post-treatment and at a 6-month follow-up. The clinical assessment consisted of a diagnostic interview and questionnaires measuring social anxiety. Following pre-treatment assessment, participants in the SAD group received nine modules of ICBT and three group-exposure sessions at the clinic, delivered over the course of 12 weeks. Post- and follow-up assessments were conducted by a clinician that had not been involved in the treatment of the participant, to reduce the risk of biased assessments. Prior to participation in the eye-tracking experiment, adolescents and parents in both the SAD and NA groups received information about the study and signed an informed consent sheet. The eye-tracking test took place in a quiet room and participants were instructed to attend to the pictures presented on the screen but no other instructions were given. The experiment lasted 20 minutes. All participants in the study received a 20 USD gift card.

### Intervention

The treatment was based on well-established CBT protocols for youth with SAD [39, 40] and included psychoeducation, functional analysis, exposure, social skills training, problem solving and relapse prevention. The treatment was a combined online and face-to-face CBT program, where nine modules were delivered through an Internet platform and three as face-to-face

**Table 1. Demographics, cognitive ability and number of successful trials in the SAD and NA groups.**

|  | SAD (N = 25) | NA (N = 22) | p value [e] |
|---|---|---|---|
| Adolescent variables |  |  |  |
| Age M (SD) | 15.1 (1.1) | 15.5 (1.2) | .28 |
| Gender (% female) | 21 (84.0) | 20 (90.9) | .67[f] |
| Cognitive ability [a] M (SD) | 108.3 (13.5) | 104.8 (13.0) | .41 |
| Social anxiety/SPAI M (SD) | 34.5 (7.6) | 8.9 (7.3) | < .001 |
| Parent variables |  |  |  |
| Level of education |  |  |  |
| Primary (%) | 11 (44.0) | 6 (27.3) | .23 |
| Higher (%) | 14 (56.0) | 16 (72.7) | .23 |
| Occupational status [b] (% employed) | 20 (80.0) | 21 (95.5) | .19[f] |
| Immigration status [c] (% immigrated) | 2 (8.0) | 0 (0.0) | .49[f] |
| Successful trials [d] |  |  |  |
| Gap–neutral faces M (SD) | 7.0 (1.2) | 7.1 (1.2) | .80 |
| Gap–happy faces M (SD) | 6.9 (1.2) | 6.9 (1.6) | .98 |
| Gap–angry faces M (SD) | 8.1 (1.3) | 7.6 (1.4) | .21 |
| Overlap–neutral faces M (SD) | 8.6 (1.5) | 8.5 (2.0) | .91 |
| Overlap–happy faces M (SD) | 8.7 (1.8) | 8.6 (1.7) | .86 |
| Overlap–angry faces M (SD) | 8.9 (1.3) | 8.6 (1.1) | .35 |

*Note.* SAD = social anxiety disorder; NA = non anxious; *M* = mean; *SD* = standard deviation.

[a] General Ability Index (GAI) based on verbal and perceptual index measured with four subtests from the Wechsler Intelligence Scale for Children 4th edition (WISC-IV, 13–16 year participants) and Wechsler Adult Intelligence Scale 4TH edition (WAIS-IV, for 17 year old participants).

[b] Number of adolescents with one or two parents with employment.

[c] Number of adolescents with one or two immigrated parents.

[d] At baseline

[e] Independent means t-test / chi-square test comparisons between groups

[f] Fisher's exact test used

group sessions. On average, adolescents completed 5.7 online modules and attended 2.1 group sessions. The group sessions were led by two therapists and with three to six participants in each group. In the online sessions, participants were instructed to read texts, watch videos and to complete various homework assignments. Parents received five separate online modules (independent from the adolescents' modules) with mainly information about SAD, strategies to reduce parental accommodation to anxiety and guidance on how to coach their adolescents when conducting exposure exercises. Parents completed on average 4.4 online modules.

## Materials

In both groups, diagnostic status was determined with the Mini International Neuropsychiatric Interview for Children and Adolescents [38] supplemented by the SAD section from the Anxiety Disorders Interview Schedule—Child Version (ADIS; [41]). In the SAD group, an experienced psychologist rated SAD severity with the Clinician Global Impression–Severity scale (CGI-S; [42]), ranging from 1 ('normal, not mentally ill') to 7 ('extremely ill'). In both the SAD and NA groups, adolescents and parents rated social anxiety symptoms with the Social Phobia and Anxiety Inventory–Child Version (SPAI-C; [43]), a 26-item scale measuring presence of social anxiety symptoms from 0 ('never or hardly ever') to 2 ('always or almost always').

To be able to control for potential differences in cognitive functioning between the SAD and NA groups, four subtests (Vocabulary, Similarities, Matrix reasoning, and Block design) from the Wechsler Intelligence Scale for Children—Fourth Edition (WISC-IV; [44]) were used to assess cognitive functioning in participants aged 13–16, and the corresponding subtest from the Wechsler Adult Intelligence Scale -Fourth Edition (WAIS-IV; [45]) were used for participants aged 17.

### Experimental paradigms

We used a modified version of the gap-overlap paradigm (e.g., [46, 47]). This paradigm has been used extensively in basic neurocognitive research on attention. In this task, participants fixate a central visual stimulus, and subsequently reorient to a stimulus that appears in the peripheral visual field (see Figs 1 and 2). Two types of trials are used. In gap trials, the central stimulus is extinguished before the peripheral stimulus appears (creating a temporal gap). The latency to reorient to the peripheral stimulus in the gap condition is therefore a relatively pure measure of the speed of target detection and visual orienting [48]. Faster orienting to threatening stimuli in the visual periphery in the gap condition can therefore serve as an index of vigilance. In overlap trials, the central stimulus remains visible when the peripheral stimulus appears (creating a temporal overlap between the two). Overlap trials generally result in longer latencies than gap trials, reflecting that visual attention has to be disengaged from the current location. Impaired disengagement from threat would therefore manifest in longer latency to disengage from threatening stimuli, whereas avoidance would manifest in faster disengagement from threatening as compared to non-threatening stimuli [49].

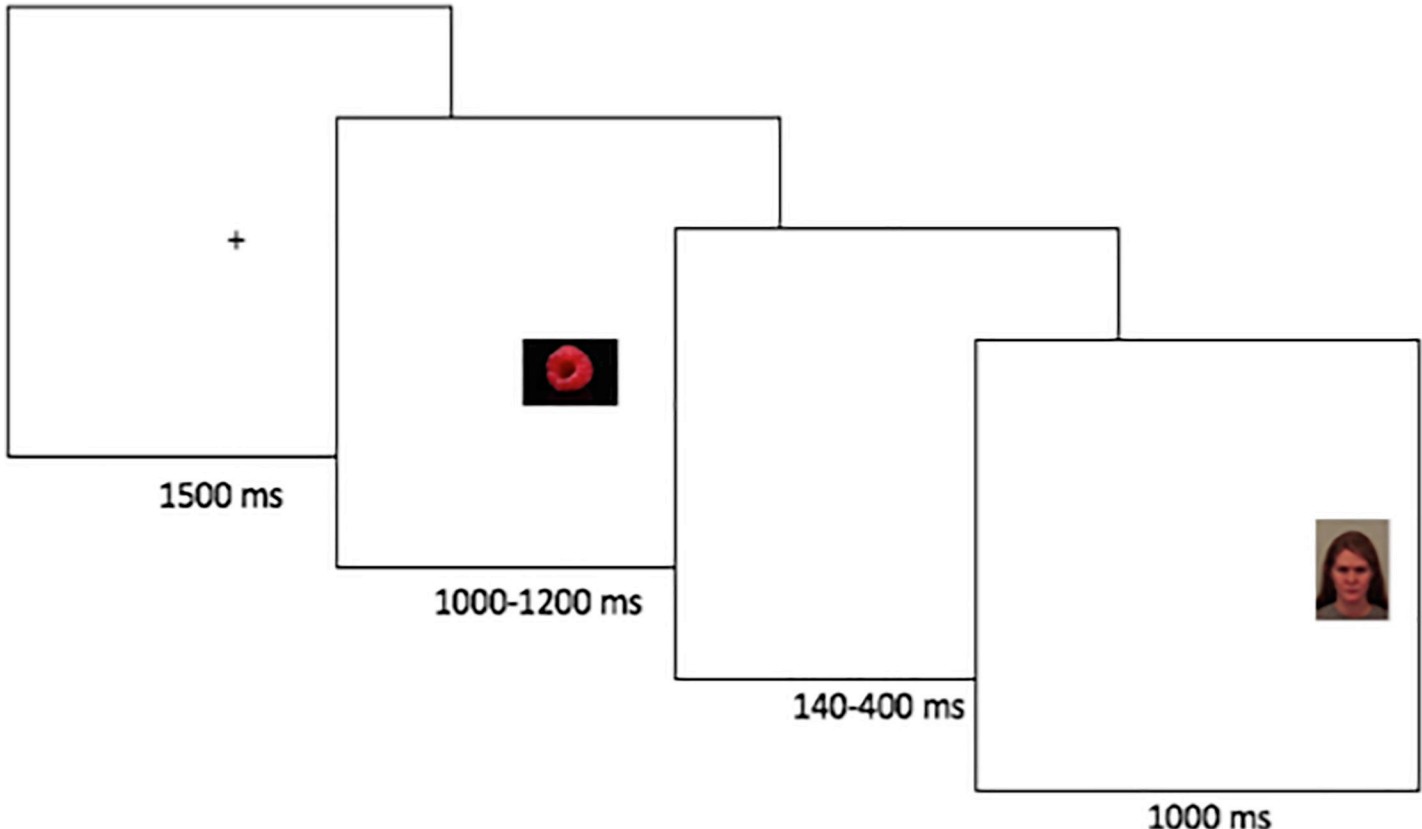

**Fig 1. Gap trial.**

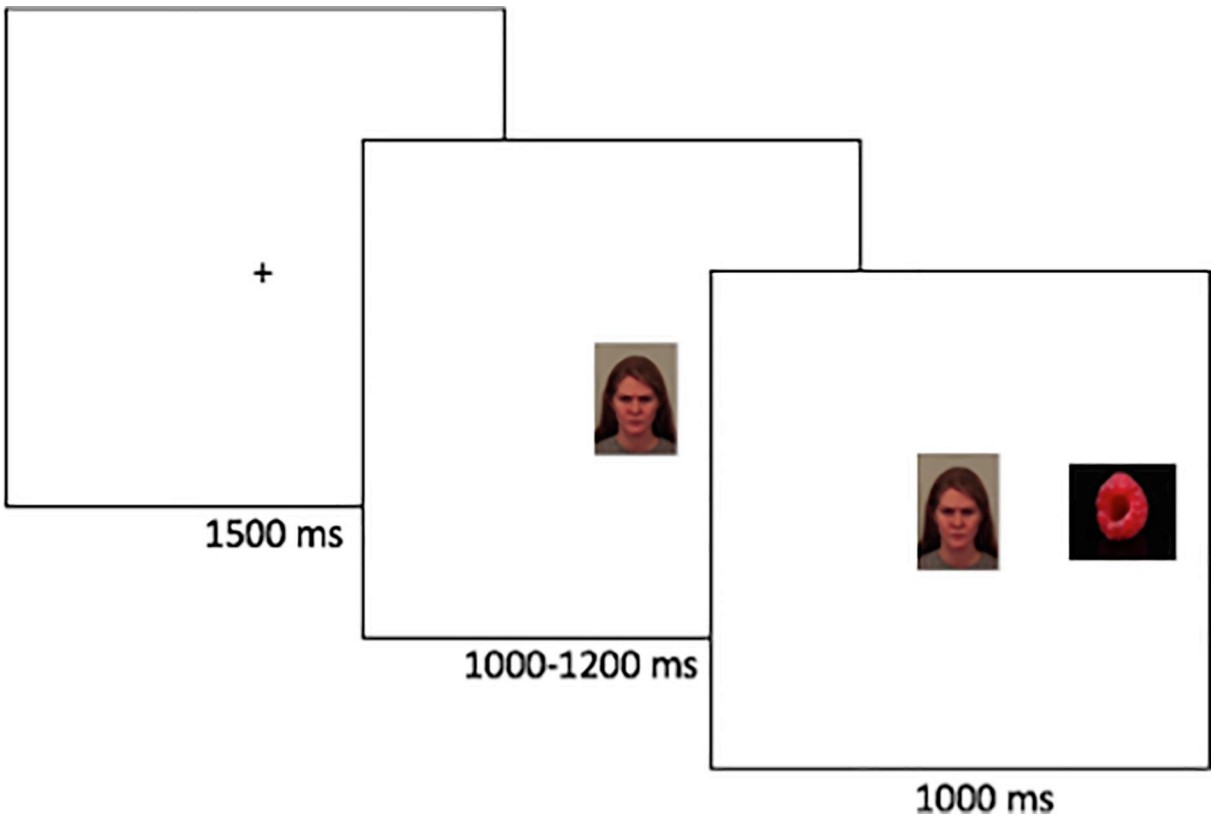

**Fig 2. Overlap trial.**

Participants completed 60 trials (30 gap- and 30 overlap trials). In overlap trials, the central stimulus was an emotional face, and the peripheral stimulus was a non-social image. On gap trials, the central stimulus was non-social, and the peripheral image was an emotional face. Facial stimuli were pictures of angry, happy and neutral human faces from the Karolinska Directed Emotional Faces (KDEF; [50]) database. Each face was shown in all three emotional states to reduce bias caused by individual person preferences, and facial emotion was balanced across conditions. Non-social stimuli were pictures of non-valenced objects such as tools and houses. To prevent predictive saccades, the gap interval varied between 140–400 ms. Gap- and overlap trials were presented interleaved in counterbalanced order.

### Recording of eye-tracking data

Gaze was recorded with a corneal reflection eye tracker (Tobii TX 120, Tobii Inc., Danderyd, Sweden) at a sample rate of 120hz. Stimuli were shown on a 17" screen. A nine-point calibration procedure took place before the actual testing began.

### Processing of eye-tracking data

A dispersion based fixation filter (Tobii fixation filter) implemented in the Tobii software, with distance and velocity thresholds set to 35 pixels was used to identify fixations. Custom scripts written in MATLAB (Mathworks, Inc) were used for all other analyses. Rectangular areas of interests (AOIs) were defined around the central and peripheral target locations. A saccade towards the peripheral target was identified if, 1) the point of gaze was outside the

**Table 2. Means and standard deviations (in milliseconds), and number of participants included in analyses, for latencies to fixate gaze on a peripherally appearing stimuli (gap trials) as well as latencies to disengage (avoidance) from a centrally located social stimuli (overlap trials).**

| Condition | SAD group Baseline | | | SAD group Post-treatments | | | SAD group 6-month follow-up | | | NA group Baseline | | |
|---|---|---|---|---|---|---|---|---|---|---|---|---|
| | *M* | *SD* | *n* | *M* | *SD* | *n* | *M* | *SD* | *n* | *M* | *SD* | n |
| Gap trials—angry | 275 | 95 | 25 | 273 | 72 | 19 | 269 | 76 | 16 | 268 | 79 | 22 |
| Gap trials—neutral | 296 | 133 | 24 | 313 | 131 | 19 | 318 | 149 | 16 | 310 | 144 | 22 |
| Gap trials—happy | 308 | 151 | 25 | 314 | 145 | 20 | 321 | 143 | 15 | 318 | 149 | 22 |
| Overlap trials—angry | 371 | 175 | 25 | 364 | 151 | 20 | 364 | 140 | 16 | 349 | 146 | 22 |
| Overlap trials—neutral | 408 | 201 | 25 | 426 | 210 | 19 | 423 | 210 | 15 | 386 | 194 | 21 |
| Overlap trials—happy | 402 | 206 | 24 | 392 | 198 | 19 | 416 | 196 | 15 | 389 | 194 | 22 |

*Note*. SAD = social anxiety disorder, NA = non anxious, *M* = means in millisecond, *SD* = standard deviations.

central AOI, and 2) reached the peripheral (target) AOI within 300 ms. The latency to initiate a first saccade towards the peripheral target was calculated from the first sample with gaze outside the central AOI. Saccades initiated earlier than 100 ms after stimulus onset were defined as anticipatory, not plausible to be reactions to the valence of the stimulus, and were disqualified from further analyses [14]. An upper limit for latency in reactive saccades was set to 1000 ms in the overlap trial and 700 ms in the gap trial. The limits for maximal latency were drawn from the 97.5th percentile in each condition. Participants contributing with less than four trials per condition were rejected from analyses in that condition (see Table 2 for number of participants included in the respective analyses).

## Statistical analyses

Data were analyzed using linear mixed effects models (LMMEs) with random intercept for participant (equal to treating multiple trials from one individual as repeated measures). The statistical significance of main and interaction effects was tested with analyses of variance (ANOVAs) with Sattherthwaite approximated degrees of freedom. An alpha level of $p < .05$ indicated statistical significance. P-values for follow-up tests are reported without correction. In addition to this traditional approach, we used Bayesian statistics to test the relative evidence for the hypotheses and null hypotheses. In contrast to a *p*-value, a Bayes Factor (BF) quantifies the relative likelihood of two statistical models–one representing the hypothesis (for example that two groups differ on a variable), and one representing the null hypothesis (that no group difference exists). A BF of 1 indicates that both models are equally well supported by the data. By convention, a BF > 3 indicates positive evidence for a model, >20 strong evidence, and BF>150 very strong evidence [51]. Bayesian statistics can therefore distinguish between a scenario where the data is inconclusive as to which hypothesis is best supported, and a scenario where the null hypothesis is best supported. This is not possible with traditional null hypothesis testing, which can aid the decision to reject the null hypothesis, but never quantify the evidence in support of it. For example, the null hypothesis is not more likely under a *p*-value of .95 than under a *p*-value of .06. However, in clinical psychiatry, it is often crucial to establish that patient and control groups *do not* differ on a variable, or that a treatment *does not* produce a change in a clinical measure. Based on recommendations from Wagenmakers [52], BFs were calculated from the Bayesian Information Criterion (BIC) values of the LMME representing the hypothesis (i.e., a model including the effect in question) and the LMME representing the null hypothesis (the most complex model not including the effect in question. The BIC weighs the maximum likelihood, the number of estimated parameters and the number of

observations, and premiers less complex models based on a larger number of observations (see [52] for details). According to Wagenmakers [52], a BF for the relative support for H0 ($BF_{01}$) can be calculated as exp (($BIC_{H}1 - BIC_{H0})$/2). For example, if $BIC_{H0} = 1.532$ and $BIC_{H1} = 1.534$, the BF in favor of H0 equals exp (1.8/2) = 2.5 (example taken from reference [52]). A BF for the relative support for H1 ($BF_{10}$) can be calculated by changing the terms to (($BIC_{H0} - BIC_{H1})$/2).

Statistical analyses were conducted in MATLAB version R2018b (Mathworks, Inc).

## Results

### Group characteristics

Table 2 displays means and standard deviations from the gap and overlap trials, between groups and across emotional conditions. Preliminary analyses revealed no statistical differences between the groups with regard to number of successfully recorded trials (see Table 1), and within the SAD group there was no effect of time (pre, post and 6-month follow-up) on number of successful trials. Due to attrition, fewer participants completed the eye-tracking experiment at post-treatment ($n = 20$) and at the 6-month follow-up ($n = 16$), as seen in Table 2. The variation in reported number of latencies included in the analyses are due to the requirement that a participant had to contribute with at least four successful trials to be included in the analyses, in each condition respectively.

### Baseline comparisons between the SAD and NA groups

**Latency to attend to peripheral emotional stimuli (vigilance—measured with gap trials).** The SAD and NA groups did not differ significantly in overall latency ($F(1, 45.29) = 0.35$, $p = .56$). A main effect of emotion was found across both groups ($F(2, 455.39) = 9.97$, $p < .001$) and follow-up tests showed that participants oriented faster to angry faces than to happy ($F(1, 340.41) = 19.52$, $p < .001$) and neutral ($F(1, 240.34) = 14.59$, $p < .001$) faces (see Fig 3). No significant difference was found between neutral and happy faces ($F(1, 632.00) = 0.70$, $p = .40$). There was no interaction between group and emotion ($F(2, 972.84) = 0.57$, $p = .57$), indicating that these effects did not differ significantly between the two groups. Both groups thus attended faster to peripherally appearing angry faces compared to happy and neutral faces, and thereby showed vigilance to threat.

**Disengagement from central emotional stimuli (avoidance–measured with overlap trials).** Similarly, there were no significant differences between the SAD and NA groups in disengagement latency ($F(1, 39.49) = 2.24$, $p = .14$). There was a main effect of emotion across the SAD and NA groups ($F(2, 1025.26) = 5.12$, $p = .006$) and post-hoc analyses showed that participants disengaged faster from angry faces than from happy ($F(1, 264.31) = 7.24$, $p < .01$) and neutral ($F(1, 490.02) = 8.54$, $p < .01$) faces (see Fig 4). No significant difference was found between neutral and happy faces ($F(1, 758.74) = 0.02$, $p = .90$) and there was no interaction between group and emotion ($F(2, 1171.72) = 0.06$, $p = .94$). In summary, both groups were thus faster to disengage from angry faces compared to happy and neutral faces, and thereby showed avoidance of threat.

### Prediction of treatment outcome in the SAD group

We tested whether vigilance and disengagement/avoidance at baseline predicted self- and clinician-rated residual symptoms at post and follow-up using a series of LMMEs with symptom level (SPAI-C or CGI-S) at baseline as a covariate. The models were initially fitted with interaction effects between emotion (angry, happy, neutral) and symptom level. No interaction effects were significant (all $p$-values >.20), and they were therefore dropped from the final models.

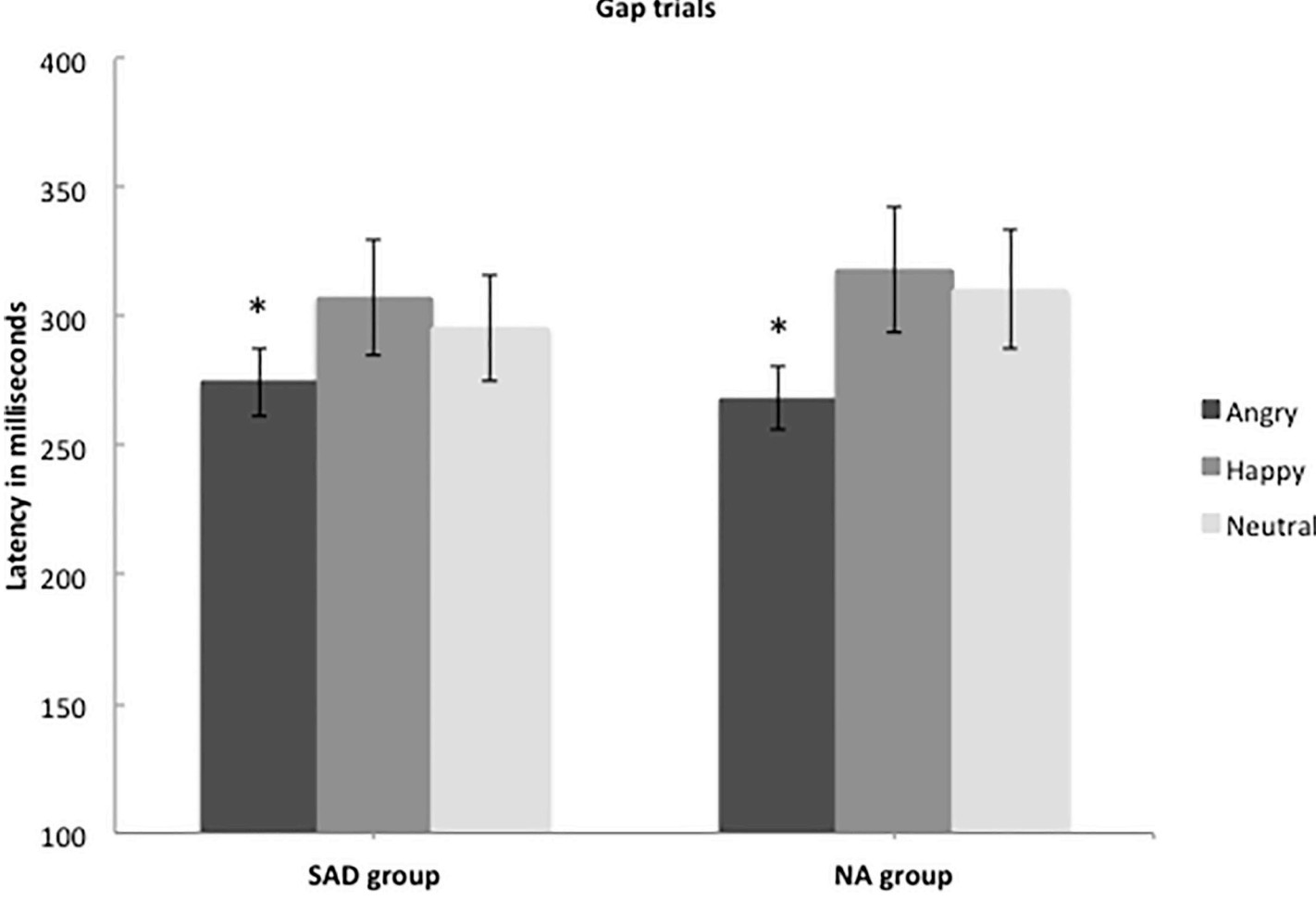

**Fig 3. Mean time in milliseconds (with 95% confidence intervals) to fixate gaze on a peripherally appearing social stimulus with different emotional valences.** For the social anxiety group ($N = 25$) and the non-anxious group ($N = 22$).

There were no main effects of baseline vigilance latency on social anxiety symptoms at post-treatment for SPAI-C ($F(1, 20.07) = 0.07$, $p = .80$) or CGI-S ($F(1, 20.41) <0.01$, $p = .95$) and neither on residual anxiety symptoms at the 6-month follow-up, measured with SPAI-C ($F(1, 16.09) = 0.05$, $p = .83$) or CGI-S ($F(1, 18.65) = 0.01$, $p = .91$).

There was a significant main effect of baseline disengagement latency on social anxiety severity at post-treatment for SPAI-C ($F(1, 13.37) = 5.28$, $p = .023$) but not for CGI-S ($F(1, 19.47) = 1.53$, $p = .21$). Furthermore, longer disengagement latency at baseline predicted higher social anxiety severity at the 6-month follow-up with SPAI-C ($F(1, 19.53) = 10.39$, $p = .004$) and with CGI-S ($F(1, 13.6) = 4.85$, $p = .045$). Those in the SAD group who were slower to disengage from pictures of faces (regardless of emotion) thus tended to have more residual social anxiety symptoms after treatment and at the 6-month follow-up, after controlling for social anxiety at baseline.

## Changes in attention after CBT

With regard to vigilance and avoidance, analyses showed that there was no main effect of time on latency to attend to peripheral stimuli ($F(2, 35.84) = 0.38$, $p = .68$) or on latency to disengage

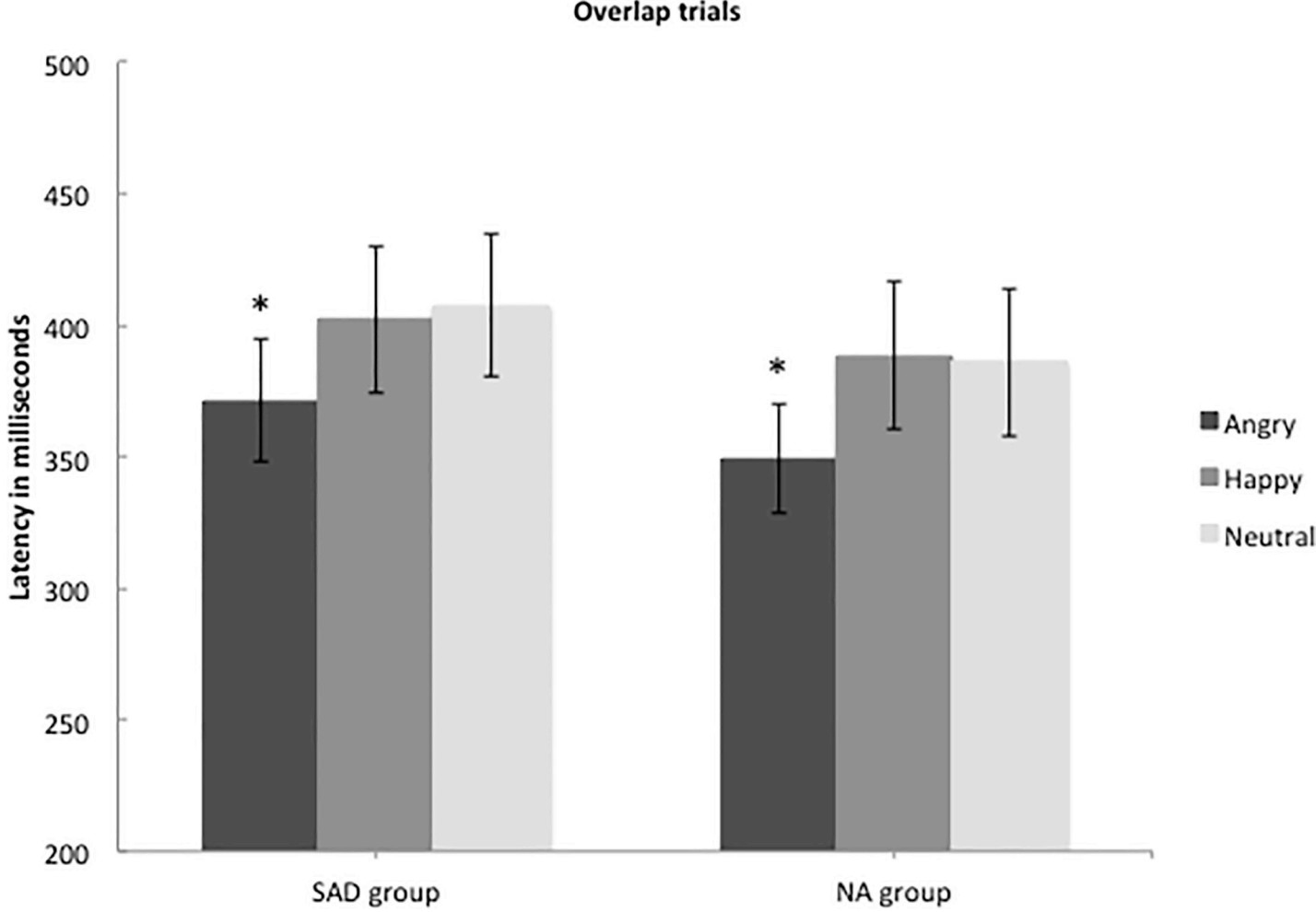

**Fig 4. Mean time in milliseconds (with 95% confidence intervals) to direct gaze away from a centrally located social stimulus, with different emotional valences.** For the social anxiety group ($N = 25$) and the non-anxious group ($N = 22$).

from central stimuli ($F(2, 35.77) = 0.13$, $p = .88$). Neither were there any interaction effects between emotion and time on vigilance ($F(4, 1322.00) = 0.55$, $p = .70$) or disengagement latency ($F(4, 1563.02) = 0.59$, $p = .67$).

### Bayesian post-hoc analyses

The above results indicated no statistically significant group differences in vigilance and disengagement latency/avoidance, as well as no significant changes in these measures over time in the SAD group. To further understand these results, Bayes factors (BF) were calculated to quantify the relative support for the hypotheses and null hypotheses related to group differences and changes over time. As noted previously, a BF can distinguish between a scenario where the null hypothesis is supported, and a scenario where the data are inconclusive. Results are presented in Table 3. As can be seen, positive to very strong evidence was found favoring the null hypothesis for most models. As an exception, the $BF_{01}$ for the main effect of group on disengagement latency was 2.19, indicating weak but inconclusive support for the null hypothesis.

**Table 3. Bayes factors indicating the relative likelihood that the observed data would occur under $H_0$ and $H_1$.**

| Comparison | Condition | $H_0$ | $BF_{01}$ | $H_1$ | $BF_{10}$ |
|---|---|---|---|---|---|
| SAD vs NA | Gap (vigilance) | No overall difference in latency | 5.71 | An overall difference in latency | 0.18 |
| SAD vs NA | Gap (vigilance) | No interaction (group*emotion) on latency | 26.48 | Interaction (group*emotion) on latency | 0.04 |
| SAD vs NA | Overlap (disengagement) | No overall difference in latency | 2.19 | An overall difference in latency | 0.46 |
| SAD vs NA | Overlap (disengagement) | No interaction (group*emotion) on latency | 44.32 | Interaction (group*emotion) on latency | 0.02 |
| SAD within group | Gap (vigilance) | No main effect of time on latency | 12.45 | A main effect of time on latency | 0.08 |
| SAD within group | Gap (vigilance) | No interaction (emotion*time) on latency | 225 | An interaction (emotion*time) on latency | 0.004 |
| SAD within group | Overlap (disengagement) | No main effect of time on latency | 21.19 | A main effect of time on latency | |
| SAD within group | Overlap (disengagement) | No interaction (emotion*time) on latency | 208 | An interaction (emotion*time) on latency | 0.005 |

*Note.* SAD = social anxiety disorder, NA = non anxious, BF = Bayes factor

## Discussion

The purpose of this study was to compare visual attention patterns in adolescents with SAD, with a non-anxious group, as well as to explore the associations between attention biases and CBT. Results demonstrated that both the SAD and the NA groups were more vigilant towards angry faces compared to neutral and happy faces, indicated by a tendency for both groups to attend faster to appearing social threats. Furthermore, both groups displayed shorter dis-engagement latencies when presented with angry faces, compared to neutral and positive faces, which we interpret as an inclination to avoid threat. Consequently, there were no signifi-cant differences between the SAD and NA groups on either vigilance or disengagement/avoid-ance but instead similar attention biases were observed in both groups. Furthermore, a general tendency to disengage slower from social stimuli in the SAD group predicted worse outcomes from ICBT. The baseline level of vigilance, however, did not predict outcome from ICBT. Lastly, vigilance and disengagement latency in the SAD group did not change over the course of treatment indicating that visual attention biases in SAD could be fairly robust traits.

We found a vigilant attention pattern in both groups, partly inconsistent with previous findings where anxious children have been shown to display more threat-related attention bias than non-anxious controls [12, 13]. Our result could have been expected in a younger sample of children, as it has been suggested that attention bias to threat is a normal phenomenon in early childhood [53] but that youth with a typical development, through cognitive maturation, learn to inhibit automatic reactivity to threat, and only those on a trajectory towards an anxiety disorder may fail to develop this control [54]. However, our results are in line with previous research using eye tracking to compare children with SAD and non-anxious controls that found similar attention patterns in both groups [18, 20]. It may be that the choice of method for capturing attention processes is important for the outcome, and our current understanding of vigilance in anxious populations stem much from studies using reaction-time measures such as the dot-probe paradigm. In the meta-analysis by Dudeney and colleagues [13] where anxious children were concluded to be more vigilant to threat than their non-anxious peers, 41 out of 44 studies employed reaction-time measures and only three used eye tracking. The snapshot nature of reaction-time measures may not be optimal for dynamic visual attention

processes such as the ones believed to be occurring during exposure to social stimuli [8]. Gaze-based measures may also capture a different aspect of attention where differences between those with and without social anxiety are less evident.

The tendency to display shorter disengagement latencies when presented angry faces, compared to neutral and positive faces, was observed in both groups. This result infers an avoidant attention pattern and combined with the finding that both groups were vigilant to threatening stimuli lends support for the vigilance-avoidance hypothesis [21]. Our results align with those found by Gamble and Rapee [25] who also noted a vigilant-avoidant attention pattern in an anxious group of children and adolescents, as well as in non-anxious controls. Other eye-tracking studies, however, have found the suggested vigilant-avoidant bias specifically in socially anxious adult populations but not in their controls (e.g., [24]) and similar group differences have been observed in studies on high-anxious adults [55, 56]. Overall, our results along with previous findings may indicate that a vigilant-avoidant bias does not differentiate anxious youths from those with a typical and non-anxious development. As this is the first eye-tracking study to investigate visual attention in a uniform group of adolescents with SAD it is, however, difficult to say whether the results are specifically related to SAD or to the adolescent age group. Future studies with wider age ranges that compare visual attention in groups with different anxiety disorders could potentially disentangle this question.

Vigilance measured before treatment did not predict the outcome of ICBT, inconsistent with findings reported by Waters and colleagues [30, 31] where selective attention to threat predicted better treatment responses. The diverging results could be explained by methodological differences between the studies (eye tracking vs. dot-probe), by the target populations (SAD vs. mixed anxiety) or by the dissimilar age groups (adolescents vs. pre-teens). Furthermore, results from the current study showed that those who disengaged attention faster from social stimuli responded better to treatment. Similar findings have been described in studies where avoidant attention biases have been linked to better treatment outcomes following CBT [33, 34, 57], although the current results indicate that more rapid disengagement from *all* sorts of social stimuli was predictive of favorable outcomes, not just avoidance of threat. It could be that CBT is particularly efficacious for those who are more avoidant of social stimuli as the treatment focuses on exposure and instructs patients to direct their attention toward other people to achieve long-term reduction of anxiety. Another interpretation of the finding that quicker disengagement predicts more favorable treatment outcome is that it reflects a role of general (i.e., not specifically social) efficacy of the brain's orienting network.

The present study did not find that visual attention patterns changed over the course of treatment, or during the 6-month follow-up period. The results are comprehensible considering that the SAD group displayed the same vigilant and avoidant attention patterns as the non-anxious group and it would not be expected that CBT could make individuals with SAD *less* vigilant or avoidant than individuals from the non-anxious population. Our results are similar to those found by Waters and colleagues [35] indicating that threat attention bias in anxious children did not change after CBT. An explanation to this could be that attention processes such as vigilance to threat and avoidance are rather fixed traits that have limited susceptibility to psychosocial interventions. Other studies have, however, found changes in attention biases following CBT [30, 36, 37] but the inconclusive findings could be related to differences in methodology, age ranges in samples or the specific forms of CBT employed across studies. In the present study, most of the CBT was delivered online and it is possible that the components targeting attention, such as exposure, was less effective than if they had been delivered face-to-face with a therapist, something that might have affected the results.

A non-significant result in traditional null-hypothesis significance testing could implicate inconclusive data as well as true support for the null hypotheses. To address this limitation, we

used Bayesian statistics to formally test the null hypotheses that groups did not differ in vigilance or disengagement latency, and that latencies did not change after CBT. In most analyses, the null hypotheses were strongly supported.

Research on the role of attention bias in the maintenance of SAD has already had implications for treatment, such as increased targeting of self-focused attention and avoidance in CBT [58]. It has also been suggested that a better understanding of attention processes in SAD could have important implications for the development of specific interventions targeting attention biases, such as attention bias modification (ABM; [6, 59]). Such interventions are thought to be able to reduce biases involved in the maintenance of SAD but the results yielded in this study showed that vigilant and avoidant attention patterns did not differentiate between adolescent with SAD and controls. This indicates that the differences between anxious and non-anxious individuals reported previously for adults [12] and partly in the youth literature [13] were not applicable to adolescents with SAD. This would possibly explain why ABM interventions for youth with SAD have been shown to have a limited effect on social anxiety [60, 61] whereas results from ABM studies on adults have been more promising (e.g., [62, 63]). Our finding that faster avoidance of social stimuli was associated with more reduction of social anxiety after ICBT shows that attention bias modulates the efficacy of treatment. This implicates that there could be subgroups of adolescents with SAD, with different attention patterns, that are more or less susceptible to CBT and that those who are more avoidant of social stimuli could be particularly amenable to exposure exercises, or other CBT components. Attention bias could thus be important to consider during initial assessment to help match patients to treatments that are most likely to lead to positive outcomes.

The study has some limitations that need to be considered. First, the sample size conveyed somewhat low statistical power, although previous similar studies on visual attention and its relation to treatment outcome have had similar sample sizes (e.g., [30, 64]). Second, the sample consisted predominantly of girls and the generalizability to a male population may therefore be limited. Lastly, as participants were recruited from a clinical trial, about half of the sample had a comorbid psychiatric disorder and although these were secondary to SAD, it may have influenced the results. It is, however, very common with comorbidity in the SAD population and a strictly homogenous SAD sample would also have had implications for the generalizability of the findings.

Despite these limitations, we believe that the study provides preliminary insights into an emerging research field in the intersection between experimental and clinical research. In conclusion, this study showed that adolescents with SAD and non-anxious adolescents were faster to detect threatening stimuli, as well as to disconnect attention from threats. Initial avoidance of social stimuli was also associated with a more favorable treatment outcome. Future studies should include multiple measures of attention, including analyses of scanpaths and pupil dilation that have been shown to capture other potentially important aspects of visual attention [65, 66]. A replication of these results in a larger sample of children and adolescents with SAD, with a wider age range, is also of interest for future research.

## Acknowledgments

We would like to express our gratitude to professor Sven Bölte and associate professor Terje Falck-Ytter for giving us the opportunity to use their facilities and eye tracker to conduct the study. We are also grateful for the possibility to conduct the study at the Clinical Research Unit (CRU) within the Child and Adolescent Psychiatry in Stockholm, Sweden.

## Author Contributions

**Conceptualization:** Jens Högström, Johan Lundin Kleberg.

**Data curation:** Jens Högström, Johan Lundin Kleberg.

**Formal analysis:** Miriam Larson Lindal, Ebba Taylor, Johan Lundin Kleberg.

**Funding acquisition:** Eva Serlachius.

**Investigation:** Jens Högström, Martina Nordh, Miriam Larson Lindal, Ebba Taylor.

**Methodology:** Jens Högström, Johan Lundin Kleberg.

**Project administration:** Jens Högström, Martina Nordh, Eva Serlachius.

**Resources:** Eva Serlachius.

**Software:** Johan Lundin Kleberg.

**Supervision:** Jens Högström, Johan Lundin Kleberg.

**Writing – original draft:** Jens Högström, Miriam Larson Lindal, Ebba Taylor, Johan Lundin Kleberg.

**Writing – review & editing:** Jens Högström, Martina Nordh, Miriam Larson Lindal, Ebba Taylor, Eva Serlachius, Johan Lundin Kleberg.

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
