## [Decision Letter · Decision Letter 0]

25 Jul 2019

PONE-D-19-18168

Visual attention to emotional faces in adolescents with social anxiety disorder receiving cognitive behavioral therapy

PLOS ONE

Dear Dr Högström,

Thank you for submitting your manuscript to PLOS ONE. After careful consideration, we feel that it has merit but does not fully meet PLOS ONE’s publication criteria as it currently stands. Therefore, we invite you to submit a revised version of the manuscript that addresses the points raised during the review process.

I think both reviewers provided excellent critiques, and they should all be addressable.

In particular, please develop on the concepts of avoidance and impaired disengagement in the introduction, as well as attention biases and vigilance, and make sure that you are consistent in the way you refer to these concepts throughout the manuscript.

Give more clarifications regarding the details of the intervention, the interviews used,  attrition rate, etc.

Please provide information about how anxiety may have influenced the results, and their interpretation.

We would appreciate receiving your revised manuscript by Sep 08 2019 11:59PM. To enhance the reproducibility of your results, we recommend that if applicable you deposit your laboratory protocols in protocols.io, where a protocol can be assigned its own identifier (DOI) such that it can be cited independently in the future. For instructions see: http://journals.plos.org/plosone/s/submission-guidelines#loc-laboratory-protocols

We look forward to receiving your revised manuscript.

Kind regards,

Nouchine Hadjikhani, MD, PhD

Academic Editor

PLOS ONE

Journal Requirements:

2)  We note that you have indicated that data from this study are available upon request. PLOS only allows data to be available upon request if there are legal or ethical restrictions on sharing data publicly. For more information on unacceptable data access restrictions, please see http://journals.plos.org/plosone/s/data-availability#loc-unacceptable-data-access-restrictions.

3) We note that Figure 1 in your submission may contain copyrighted images. All PLOS content is published under the Creative Commons Attribution License (CC BY 4.0), which means that the manuscript, images, and Supporting Information files will be freely available online, and any third party is permitted to access, download, copy, distribute, and use these materials in any way, even commercially, with proper attribution. For more information, see our copyright guidelines: http://journals.plos.org/plosone/s/licenses-and-copyright.

i) You may seek permission from the original copyright holder of Figure(s) [#] to publish the content specifically under the CC BY 4.0 license.

ii).    If you are unable to obtain permission from the original copyright holder to publish these figures under the CC BY 4.0 license or if the copyright holder’s requirements are incompatible with the CC BY 4.0 license, please either i) remove the figure or ii) supply a replacement figure that complies with the CC BY 4.0 license. Please check copyright information on all replacement figures and update the figure caption with source information. If applicable, please specify in the figure caption text when a figure is similar but not identical to the original image and is therefore for illustrative purposes only.

Reviewers' comments:

Reviewer's Responses to Questions

**Comments to the Author**

1. Is the manuscript technically sound, and do the data support the conclusions?

Reviewer #1: Yes

Reviewer #2: Yes

2. Has the statistical analysis been performed appropriately and rigorously? 

Reviewer #1: Yes

Reviewer #2: Yes

3. Have the authors made all data underlying the findings in their manuscript fully available?

Reviewer #1: Yes

Reviewer #2: No

4. Is the manuscript presented in an intelligible fashion and written in standard English?

Reviewer #1: Yes

Reviewer #2: Yes

5. Review Comments to the Author

Reviewer #1: PONE-D-19-18168

The present manuscript describes a single study examining attention biases in adolescents diagnosed with SAD and a Non-Anxiety control group (NA) as well as the effect of the internet-based cognitive behavioural therapy on the attention biases in the clinical sample (pre-treatment, post- treatment and at a 6-month follow up). Attention biases examined by the study were vigilance to and disengagement from threatening stimuli as measured using eye tracking. To get at vigilance, the authors measured latency to reorient to the peripheral stimulus (emotional face) from the previously presented central stimulus (object) in the gap trials. Faster orienting to threatening stimulus during gap trials means better vigilance. To get at disengagement, the authors measured latency to orient to the peripheral stimulus (object) when presented along with the central stimulus (emotional face) in the overlap trials. Longer latency to disengage from the emotional face indexes impaired disengagement and faster disengagement would mean avoidance from threatening stimuli toward object.

The results were:

- No group differences in the gap trials with both groups orienting faster to angry than happy or neutral faces

- No group differences in overlap trials both groups disengage faster from angry faces compared to happy and neutral

- Those with SAD who had impaired disengagement at baseline had more social anxiety symptoms as measured by SPAI-C (POST AND 6 MONTHS) AND and CGI-S (AT 6MO), and presumably those with a more avoidant visual attention to faces resulted in less anxiety symptoms.

Overall, I appreciated reading this paper because it separated the types of visual attention processes using an objective measure such as eye gaze. However, it was a bit difficult to find it very exciting since the results found no differences between the groups nor any significant changes after treatment. With that said, I do think the manuscript needs revision prior to publication. Based on my reading, I found several points that need adjusting, explanation and revision, some more major than others. I will point these out in the ‘chronological’ order of the manuscript.

1. In the abstract, it would be helpful for a reader to get a defining sentence of what types of attentional biases that are being examined. The authors say so on line 55, but I think a sentence stating it explicitly might be best.

2. Throughout the introduction I was confused by the relationship between vigilance avoidance and impaired disengagement. I was especially confused because avoidance and impaired disengagement seem to be two opposing processes (either short or long latency) but the actual limits are not clear – seems to be dependent on what is being measured in comparison. But because the authors focus their study on examining vigilance and impaired disengagement specifically, I would try to focus the introduction on those two concepts, with an extended information about the implications of impaired disengagement (what does that mean for those with anxiety disorders?). I understand that the reason why the authors talk about vigilance and avoidance is because of the past research linked those two together, but I think it tends to be confusing to the reader. Finally, the authors also refer to impaired disengagement differently using the term ‘disengagement/avoidance’ only in the discussion, but not anywhere else. Please be consistent.

3. Table 1 - The authors should consider whether the differences in a ADIS, MINI and SPAI should be added.

4. The ‘Intervention’ section on line to 34 page 11 is a bit unclear. Specifically, I am unclear on how many sessions the participants have had and whether the parental sessions were in addition to the nine participant sessions? Please clarify. Also, did all participants attend/complete all the parts of the treatment?

5. Results section - under the ‘Comparison between the SAD and NA groups’ Is it SAD baseline vs. NA?

6. Table 2 shows the number of participants in the SAD group baseline, post treatment and follow up. These numbers change significantly (n = 25 at baseline vs. n = 16 at 6-month follow up) Authors need to clear up why these numbers change. Attrition rate? Incomplete treatment?

7. Is there a way to see how many trials were completed by each group on average?

8. On page 19 line 417 the authors say are the latency was 2.2 but the table indicates 2.19

9. Throughout the paper, the authors go back and forth between saying stimulus is threatening (which I’m assuming means angry faces) to suggesting ALL social stimulus is being avoided (presumably because there were no main effect of emotion). It would be beneficial if the authors discussed more why all social stimuli (happy and neutral) faces could be regarded as potentially threatening to people with SAD or even adolescents with NA?

10. A big thumbs up for lines 520-525 pg. 24 of the Discussion!

Reviewer #2: This study investigates the role of attention bias in youth with social anxiety disorder (SAD), and the association between attention bias and outcome following cognitive behavioral therapy (CBT) intervention. More specifically, the authors measured attention bias through vigilance to faces, as measured by latency to disengage attention from these social stimuli. First, attention bias was compared (at baseline) between non-anxious children and those with SAD; the authors found no group differences in disengagement from faces depicting various emotions. Second, the authors examined the relationship between attention disengagement from emotion faces with CBT outcome, and changes in attention bias following the CBT intervention. Results indicated that youth with SAD who showed a faster disengagement pattern had less anxiety following the intervention. There were no changes in attention bias following the intervention. The authors should address the following issues:

Introduction:

1. The introduction would benefit from more comprehensive review of attention biases in anxiety, and SAD in particular, that extends beyond experimental findings. This would help set the stage for the importance of the study.

2. The introduction would benefit from some restructuring in terms of order of presented ideas. For example, it seems that review of attention bias across anxiety disorders (starting at line 104) should come before more specific review of this in SAD (starting at line 87).

3. What is the hypothesis for why vigilance to threat is moderated by age (lines 108-111)? May be helpful to discuss this in light of findings in Discussion.

4. Please cite definition of disengagement (lines 114-115). How does disengagement contribute to the development of SAD? A more thorough review is needed as this is the central conceptual framework of this study. Why is it important to assess all three aspects of attention; do you do this in the present study?

5. It seems that the authors use the term “vigilance” to mean faster latency to shift, and “avoidance” to mean slower disengagement, though this is not always clear/consistent in use of language. Consistency in how these concepts are referenced would be helpful throughout the manuscript (including in results). I find myself going back and forth through manuscript to “check” on what concept/process the authors are referencing.

6. “Children in the non-anxious group did not show any attention bias in any direction (line 128-129) – does this mean there were no differences in disengagement to faces by emotion type?

7. What is meant by “later stages of processing” (line 146)?

8. Please review what aspects of CBT, and especially ICBT, address attention bias at a basic attentional and emotional processing level (this would be helpful in Intro and again discussed in light of findings in the Discussion).

9. Why is vigilance toward threat predictive of greater symptom reduction (line 162-163)?

10. The authors may consider rephrasing “whether it is more beneficial” (line 169) to summarizing the link between the described bias and positive intervention outcome.

Method:

1. The authors state that all SAD participants fulfilled DSM-5 criteria for a principal diagnosis of SAD in Participants section. May be helpful for clarification to include the name of the interview and that it was used for both groups (so the reader does not have to search for this info in Materials section). Relatedly, why were two structured interviews used in the SAD group used?

2. The authors have designed the experimental testing of attention bias through use of a gap-overlap task. It is unclear if the gap and overlap trials were run together or in separate blocks.

3. Additional citations are needed in the Experimental Paradigm section (e.g., lines 267-269; 271-275).

Discussion

1. The Discussion section is opened with the “main purpose” of the study, which leaves out half of the research questions addressed.

2. Are there any differences in emotion stimuli used between this study and others in the literature? Could that have an impact on findings of no group differences in both gap and overlap tasks?

3. Is there some sort of developmental process in adolescence occurring that could account for lack of group differences between NA and SAD, especially in light of finding of differences during other developmental periods (childhood, adulthood)?

4. The authors don’t provide information on co-occurring anxiety diagnoses and this would be helpful. Could this impact the interpretation of results? For example, could co-occurring diagnosis (anxiety or otherwise) impact why there are no findings with regard prediction of treatment outcome or to changes in attention processes following treatment? Please address in Method/Discussion.

5. What could the potential impact be of ICBT intervention, versus more traditional in-person CBT, be on findings? For example, is there less emphasis on exposure, and could this impact the findings of no change in attentional processes post-treatment?

Minor points to address:

1. The sentence, “One of the advantages…” (lines 101-103) is not clear. Eye tracking does not measure shifts “over time” as trials are averaged. Please clarify what is meant here. The sentence, “This practice has been established…” (lines 120-121) is not clear. Please rephrase to clarify point.

2. Please separate paragraph starting at line 118, starting with “ There are, however, few….

3. There are several typos and manuscript needs to be proof read; for example, line 161 is missing youth “with” SAD.

4. Please identify that the numbers reported in Table 2 are in milliseconds.

5. Figures 3 & 4: Please consider adding indicators of significant results where they exist.

6. PLOS authors have the option to publish the peer review history of their article (what does this mean?). If published, this will include your full peer review and any attached files.

Reviewer #1: No

Reviewer #2: No

---

## [Author Response · Author response to Decision Letter 0]

29 Aug 2019

Response to reviewers

Dear Dr Hadjikhani,

Thank you for allowing us to revise and resubmit the manuscript “Visual attention to emotional faces in adolescents with social anxiety disorder receiving cognitive behavioral therapy” (PONE-D-19-18168). We have done our best to address the critique and feedback received from the reviewers as well as from you as the editor. We felt that the questions raised were excellent and pointed out a number of important concerns, and the manuscript is now substantially improved in our eyes.

Best regards

Jens Högström

PhD, Lic. Psychologist 

Karolinska Institutet

Child and Adolescent Psychiatry

Clinical Research Unit

Gävlegatan 22 | 113 30 Stockholm

E-mail: jens.hogstrom@ki.se

Phone: 0046 8 514 522 08

Points raised by the academic editor

1. In particular, please develop on the concepts of avoidance and impaired disengagement in the introduction, as well as attention biases and vigilance, and make sure that you are consistent in the way you refer to these concepts throughout the manuscript.

Reply: We have now gone through the manuscript and defined these concepts better as well as made sure to be consistent when referring to them. Please see our replies to the reviewers regarding this issue.

2. Give more clarifications regarding the details of the intervention, the interviews used, attrition rate, etc.

Reply: This has now been done.

3. Please provide information about how anxiety may have influenced the results, and their interpretation.

Reply: We have now added information in the method and discussion sections where we address this question.

4. Restrictions regarding sharing of data.

Reply: The data set contains sensitive clinical data, concerning diagnostic status of participants, how they respond to treatment as well as related information about their mental health. We are restricted by the Regional ethical review board in Stockholm to share this data publicly, but upon request we will be able to share data concerning visual attention and other variables that does not immediately give away clinical features of the study participants. The data set can be obtained upon request by e-mailing the corresponding author (jens.hogstrom@ki.se) or the Clinical Research Unit within the Child and Adolescent Psychiatry in Stockholm (bupkfe.slso@sll.se).

5. “Figure 1 in your submission may contain copyrighted images. All PLOS content is published under the Creative Commons Attribution License (CC BY 4.0), which means that the manuscript, images, and Supporting Information files will be freely available online, and any third party is permitted to access, download, copy, distribute, and use these materials in any way, even commercially, with proper attribution. We require you to either (1) present written permission from the copyright holder to publish these figures specifically under the CC BY 4.0 license, or (2) remove the figures from your submission:

Reply: The KDEF is a very widely used standardized set of facial stimuli. The creators of the stimulus set have given a general permission to publish sample images, or experimental stimuli containing these images, in scientific journals. This statement can be found at: 

http://kdef.se/home/using%20and%20publishing%20kdef%20and%20akdef.html

Reviewers' comments:

 Reviewer #1

The present manuscript describes a single study examining attention biases in adolescents diagnosed with SAD and a Non-Anxiety control group (NA) as well as the effect of the internet-based cognitive behavioural therapy on the attention biases in the clinical sample (pre-treatment, post- treatment and at a 6-month follow up). Attention biases examined by the study were vigilance to and disengagement from threatening stimuli as measured using eye tracking. To get at vigilance, the authors measured latency to reorient to the peripheral stimulus (emotional face) from the previously presented central stimulus (object) in the gap trials. Faster orienting to threatening stimulus during gap trials means better vigilance. To get at disengagement, the authors measured latency to orient to the peripheral stimulus (object) when presented along with the central stimulus (emotional face) in the overlap trials. Longer latency to disengage from the emotional face indexes impaired disengagement and faster disengagement would mean avoidance from threatening stimuli toward object.

The results were:

- No group differences in the gap trials with both groups orienting faster to angry than happy or neutral faces

- No group differences in overlap trials both groups disengage faster from angry faces compared to happy and neutral

- Those with SAD who had impaired disengagement at baseline had more social anxiety symptoms as measured by SPAI-C (POST AND 6 MONTHS) AND and CGI-S (AT 6MO), and presumably those with a more avoidant visual attention to faces resulted in less anxiety symptoms.

Overall, I appreciated reading this paper because it separated the types of visual attention processes using an objective measure such as eye gaze. However, it was a bit difficult to find it very exciting since the results found no differences between the groups nor any significant changes after treatment. With that said, I do think the manuscript needs revision prior to publication. Based on my reading, I found several points that need adjusting, explanation and revision, some more major than others. I will point these out in the ‘chronological’ order of the manuscript.

1. In the abstract, it would be helpful for a reader to get a defining sentence of what types of attentional biases that are being examined. The authors say so on line 55, but I think a sentence stating it explicitly might be best.

Reply: We agree with the reviewer that this could have been stated more explicitly and have now added the following information to the abstract:

“Vigilance was operationalized as the time it took to relocate the gaze from a central position to a peripherally appearing social stimulus. The latency to disengage from a centrally located social stimulus, when a non-social stimulus appeared in the periphery, was used as a proxy for avoidance.”

2. Throughout the introduction I was confused by the relationship between vigilance avoidance and impaired disengagement. I was especially confused because avoidance and impaired disengagement seem to be two opposing processes (either short or long latency) but the actual limits are not clear – seems to be dependent on what is being measured in comparison. But because the authors focus their study on examining vigilance and impaired disengagement specifically, I would try to focus the introduction on those two concepts, with an extended information about the implications of impaired disengagement (what does that mean for those with anxiety disorders?). I understand that the reason why the authors talk about vigilance and avoidance is because of the past research linked those two together, but I think it tends to be confusing to the reader. Finally, the authors also refer to impaired disengagement differently using the term ‘disengagement/avoidance’ only in the discussion, but not anywhere else. Please be consistent.

Reply: This is a very good point and we have now changed the part of the introduction where we describe vigilance, avoidance and impaired disengagement, detailing how these three attentional processes are commonly operationalized in eye-tracking studies. The reviewer is correct in pointing out that in this study we are looking at disengagement latency toward different emotional stimuli as potentially relatively longer (indicating impaired disengagement) or relatively shorter (indicating avoidance), and this has now hopefully been addressed in the introduction as well as in the method section. As the reviewer also points out, the past literature focuses heavily on vigilance and avoidance and we are attempting to align with that literature, while at the same time aiming to describe that we have conceptualized avoidance somewhat differently. Hopefully this is now clearer to the reader. We have also gone through the entire manuscript to ensure that we are consistent when we talk about disengagement latency and avoidance. Lastly, we have upon the suggestion extended the information about the implications of impaired disengagement for someone with an anxiety disorder.

Introduction (page 6, changes in italics):

”In eye-tracking studies, vigilance (biased attention toward threat) is often defined as a proneness to orient faster, or more often, to threatening stimuli, as opposed to non-threatening stimuli. Avoidance (biased attention away from threat) is conversely commonly defined as an inclination to saccade faster, or more frequently, to non-threatening stimuli compared to threatening stimuli (8). Another process of presumed importance for maintenance of anxiety is attentional disengagement. In eye-tracking studies, disengagement can be conceptualized as the latency to detach the gaze from a stimulus (e.g., 14). A relatively longer latency to disengage from a fixated stimulus indicates impaired disengagement whereas shorter latency infers avoidance of, for instance, a threatening stimulus. Difficulty with disengagement has been shown to contribute to social anxiety in adults and the ability to disengage adequately from threats is thought to prevent an individual from ruminating too much on negative aspects of the surrounding (15, 16). Attentional avoidance of social threat cues are also believed to exacerbate anxiety in youth, but through a tendency to miss out on important social information and thereby risking an exaggeration of the perceived threat (5).”

Methods (page 14):

“Overlap trials generally result in longer latencies than gap trials, reflecting that visual attention has to be disengaged from the current location. Impaired disengagement from threat would therefore manifest in longer latency to disengage from threatening stimuli (impaired disengagement), whereas avoidance would manifest in faster disengagement from threatening as compared to non-threatening stimuli.”

3. Table 1 - The authors should consider whether the differences in a ADIS, MINI and SPAI should be added.

Reply: We have now added the means and standard deviations of SPAI for the two groups to table 1. However, we think it may not be necessary to describe that all participants in the SAD group were diagnosed with social anxiety disorder and that no one in the NA group had social anxiety disorder (according to MINI/ADIS) as this is rather clearly outlined in the Participants paragraph and should neither require a statistical comparison between the groups. 

4. The ‘Intervention’ section on line to 34 page 11 is a bit unclear. Specifically, I am unclear on how many sessions the participants have had and whether the parental sessions were in addition to the nine participant sessions? Please clarify. Also, did all participants attend/complete all the parts of the treatment?

Reply: We agree that this was not clear and information about the number of completed online modules and group sessions has now been added, for both adolescents and parents. We have also clarified that the five parental modules were delivered on top of / separately from the 9 adolescent modules (see page 13 in the manuscript).

5. Results section - under the ‘Comparison between the SAD and NA groups’ Is it SAD baseline vs. NA?

Reply: Yes that is correct and to make this more clear we have now changed the headline to:

“Baseline comparison between the SAD and NA group”

6. Table 2 shows the number of participants in the SAD group baseline, post treatment and follow up. These numbers change significantly (n = 25 at baseline vs. n = 16 at 6-month follow up) Authors need to clear up why these numbers change. Attrition rate? Incomplete treatment?

Reply: Yes, the varying numbers are du to attrition at post-treatment and the 6-month follow-up, as well as due to the required minimum number of successful trials per participant per condition. We now hope to have made this more unambiguous by adding the following text to the first part of the Results section (page 17):

”Due to attrition, fewer participants completed the eye-tracking experiment at post-treatment (n=20) and at the 6-month follow-up (n=16), as seen in Table 2. The variation in reported number of latencies included in the analyses are due to the requirement that a participant had to contribute with at least four successful trials to be included in the analyses, in each condition respectively.”

7. Is there a way to see how many trials were completed by each group on average?

Reply: In table 1 (page 11) we report the average number of trials that were successful per condition (Gap – neutral faces, Gap – happy faces etc). The means range from 6.9-8.9 trials per condition (out of 10 possible trials). All in all, participants completed 60 trials (30 Gap and 30 Overlap trials), as described under Experimental paradigms on page 13, but after processing of the eye-tracking data, some of these trials were found to not have been recorded properly, or were for other reasons not considered successful. If the reviewer feels that this data should be presented otherwise, such as with averages per group notwithstanding condition, we are happy to do so.

8. On page 19 line 417 the authors say are the latency was 2.2 but the table indicates 2.19

Reply: Thank you for pointing this out, we have now changed the figure in the text to “2.19”, to correspond to the more exact figure in the table.

9. Throughout the paper, the authors go back and forth between saying stimulus is threatening (which I’m assuming means angry faces) to suggesting ALL social stimulus is being avoided (presumably because there were no main effect of emotion). It would be beneficial if the authors discussed more why all social stimuli (happy and neutral) faces could be regarded as potentially threatening to people with SAD or even adolescents with NA?

Reply: We are not entirely sure that we understand the reviewer here, as we did find main effects of emotion in gap- as well as overlap trials, as described in the Results section (page 19):

“A main effect of emotion was found across both groups (F(2,455.39)=9.97, p<.001) and follow-up tests showed that participants oriented faster to angry faces than to happy (F(1,340.41)=19.52, p<.001) and neutral (F(1,240.34)=14.59, p<.001) faces”

and

“There was a main effect of emotion across the SAD and NA groups (F(2,1025.26)=5.12, p=.006) and post-hoc analyses showed that participants disengaged faster from angry faces than from happy (F(1,264.31)=7.24, p<.01) and neutral (F(1,490.02)=8.54, p<.01) faces” 

Perhaps the reviewer is referring to other studies that are described in the introduction, for instance the Chen & Clarke (2017) review where they are (in turn) referring to studies where vigilant and avoidant attention biases have been found in relation to threatening as well as other emotional stimuli (e.g., happy faces). As we did find main effects of emotion in the between-group comparisons in the current trial (as expected), we thought it not motivated to go into a discussion about why other trials may not have found effects of different emotions. We do however believe that the introduction would benefit from a motivation to why we chose to include positive stimuli (happy faces) in the gap- and overlap trials. We have therefore added the following text to the last part of the introduction (pages 9-10):

”The social stimuli consisted of pictures of angry, neutral as well as happy faces, as previous studies have indicated that not only threatening but also positive emotional stimuli can be negatively perceived in SAD (8).”

10. A big thumbs up for lines 520-525 pg. 24 of the Discussion!

Reply: Thanks!

Reviewer #2: This study investigates the role of attention bias in youth with social anxiety disorder (SAD), and the association between attention bias and outcome following cognitive behavioral therapy (CBT) intervention. More specifically, the authors measured attention bias through vigilance to faces, as measured by latency to disengage attention from these social stimuli. First, attention bias was compared (at baseline) between non-anxious children and those with SAD; the authors found no group differences in disengagement from faces depicting various emotions. Second, the authors examined the relationship between attention disengagement from emotion faces with CBT outcome, and changes in attention bias following the CBT intervention. Results indicated that youth with SAD who showed a faster disengagement pattern had less anxiety following the intervention. There were no changes in attention bias following the intervention. The authors should address the following issues:

Introduction:

1. The introduction would benefit from more comprehensive review of attention biases in anxiety, and SAD in particular, that extends beyond experimental findings. This would help set the stage for the importance of the study.

Reply: We welcome this suggestion and agree that the manuscript could be improved by including a somewhat wider context to attention biases in SAD in the introduction. We have added the following information (page 4):

”Proposed etiological and maintaining factors of social anxiety include different cognitive biases, such as tendencies to interpret social situations negatively, excessive self-focused scrutiny and a propensity to make inaccurate inferences from internal information (such as anxiety) about how one is perceived by others (2)”

and (further down on page 4):

“Findings from experimental as well as non-experimental studies point to the importance of these factors in social anxiety. Self-report studies assessing focus of attention have, for instance, shown that there is a link between self-focused attention and presence of SAD but the validity of assessing direction of attention with questionnaires in youth populations has been questioned (2). Therefore, the majority of studies in the attention field have used experimental designs to capture how individuals with SAD observe the outside world.”

We would have liked to elaborate more on this topic but we feel that a more comprehensive review might be outside the scope of the conducted study.

2. The introduction would benefit from some restructuring in terms of order of presented ideas. For example, it seems that review of attention bias across anxiety disorders (starting at line 104) should come before more specific review of this in SAD (starting at line 87).

Reply: What starts on line 87 in the original manuscript (line 96 in the revised manuscript) is merely a description of the problems with using the dot-probe paradigm to measure attention biases. The review of the SAD literature starts on line 118 in the original manuscript (line 144 in the revised manuscript). To hopefully clarify this we have now removed “and SAD” from the original line 87:

“The majority of studies on attention biases in anxiety disorders and SAD have been conducted with the dot-probe paradigm.”

We acknowledge that we do refer to studies including youth populations with a mixture of anxiety disorder, further down in the introduction, but only in instances where the samples include children/adolescents with SAD (which we have tried to point out). We have thought about other ways to structure the introduction but not really found a more logical way.

3. What is the hypothesis for why vigilance to threat is moderated by age (lines 108-111)? May be helpful to discuss this in light of findings in Discussion.

Reply: The hypothesis for why age could be a moderator of vigilance to threat is described in Field and Lester (2010) where younger children are suggested to generally respond to information from the surrounding in a bottom-up and valence-driven way, more than older children/adolescents/adults. As executive functioning develops with age, normally developed children learn to inhibit automatic responding and subsequently become less reactive to threatening stimuli, whereas those with anxiety disorders maintain the biased attention towards threat. 

 We attempt to discuss this, in the discussion, on page 23:

“Our result could have been expected in a younger sample of children, as it has been suggested that attention bias to threat is a normal phenomenon in early childhood (53) but that youth with a typical development, through cognitive maturation, learn to inhibit automatic reactivity to threat, and only those on a trajectory towards an anxiety disorder may fail to develop this control (54).”

To prepare for this discussion we have now added the following information in the introduction, on page 5-6:

“However, for children, this vigilance to threat seems to be present in non-anxious individuals as well, and the effect has been shown to be moderated by age across studies (13). This meta-analysis showed that it was more likely to find attention bias differences between anxious and non-anxious groups in older youth samples, than in samples with younger children. The reason for this is thought to be related to a general tendency for younger children to respond to information from the surrounding in a bottom-up and valence-driven way. With maturation though, normally developed children learn to inhibit automatic responding and gradually become less reactive to threatening stimuli, whereas those with an anxiety disorder looks to maintain the biased attention towards threat (13).

4. Please cite definition of disengagement (lines 114-115). How does disengagement contribute to the development of SAD? A more thorough review is needed as this is the central conceptual framework of this study. Why is it important to assess all three aspects of attention; do you do this in the present study?

Reply: Thank you for pointing this out, we agree that this was not evident in the previous version of the manuscript. Reviewer 1 had a very similar comment and we have changed and added information regarding this in the introduction (see also response to Reviewer 1, comment 2). The following definitions, link between disengagement difficulties and SAD, as well as clarifications about what we measure in this study have now been added:

Introduction, page 6:

”In eye-tracking studies, vigilance (biased attention toward threat) is often defined as a proneness to orient faster, or more often, to threatening stimuli, as opposed to non-threatening stimuli. Avoidance (biased attention away from threat) is conversely commonly defined as an inclination to saccade faster, or more frequently, to non-threatening stimuli compared to threatening stimuli (8). Another process of presumed importance for maintenance of anxiety is attentional disengagement. In eye-tracking studies, disengagement can be conceptualized as the latency to detach the gaze from a stimulus (e.g., 14). A relatively longer latency to disengage from a fixated stimulus indicates impaired disengagement whereas shorter latency infers avoidance of, for instance, a threatening stimulus. Difficulty with disengagement has been shown to contribute to social anxiety in adults and the ability to disengage adequately from threats is thought to prevent an individual from ruminating too much on negative aspects of the surrounding (15, 16). Attentional avoidance of social threat cues are also believed to exacerbate anxiety in youth, but through a tendency to miss out on important social information and thereby risking an exaggeration of the perceived threat (5).”

See also Methods (page 14):

“In gap trials, the central stimulus is extinguished before the peripheral stimulus appears (creating a temporal gap). The latency to reorient to the peripheral stimulus in the gap condition is therefore a relatively pure measure of the speed of target detection and visual orienting (48). Faster orienting to threatening stimuli in the visual periphery in the gap condition can therefore serve as an index of vigilance. In overlap trials, the central stimulus remains visible when the peripheral stimulus appears (creating a temporal overlap between the two). Overlap trials generally result in longer latencies than gap trials, reflecting that visual attention has to be disengaged from the current location. Impaired disengagement from threat would therefore manifest in longer latency to disengage from threatening stimuli (impaired disengagement), whereas avoidance would manifest in faster disengagement from threatening as compared to non-threatening stimuli (49).”

5. It seems that the authors use the term “vigilance” to mean faster latency to shift, and “avoidance” to mean slower disengagement, though this is not always clear/consistent in use of language. Consistency in how these concepts are referenced would be helpful throughout the manuscript (including in results). I find myself going back and forth through manuscript to “check” on what concept/process the authors are referencing.

Reply: We have now gone through the manuscript and made this more clear and consistent.

6. “Children in the non-anxious group did not show any attention bias in any direction (line 128-129) – does this mean there were no differences in disengagement to faces by emotion type?

Reply: In this study, the dot-probe paradigm was used to determine the likelihood that participants attended to pictures of angry, happy and neutral faces when they were presented in pairs (angry-neutral, happy-neutral or neutral-neutral). That is, they did not measure disengagement but rather vigilance and avoidance. To clarify the finding from this study, we have bow added the following information (page 7):

“Children in the non-anxious control group did not show any attention bias in either direction, i.e., they were as likely to direct their gaze toward angry as well as to neutral faces when they were presented in pairs (19).

7. What is meant by “later stages of processing” (line 146)?

Reply: Typically, later stages of processing means >1000ms but this seems to vary a bit between studies. In the study by Gamble & Rapee (2009), for instance, they look at attentional deployment during 500ms presentations of stimuli and compare with 3000ms presentations. The first is referred to as “initial orienting” the other as “sustained processing” and correspond to what we mean by “initial” and “later” stages of processing. To clarify we have now added information about this (page 8):

“although there is no evident time-course across studies as vigilance and avoidance have been found at both initial (0-1000 ms) and later (>1000 ms) stages of processing.”

8. Please review what aspects of CBT, and especially ICBT, address attention bias at a basic attentional and emotional processing level (this would be helpful in Intro and again discussed in light of findings in the Discussion).

Reply: We agree that this should be highlighted more and have now added the following information to the introduction (page 9):

”CBT includes components that target different aspects of attention, e.g., redirection of attention toward threat (during exposure) and cognitive restructuring that aim to change the appraisal of threats in the environment. Studying if attention is modifiable by treatment is therefore important as it may inform us about whether attention bias in SAD is a stable trait or if it could be targeted in treatment as a means to reduce anxiety symptoms (35).”

And to the discussion (pages 25-26):

“Other studies have, however, found changes in attention biases following CBT (30, 36, 37) but the inconclusive findings could be related to differences in methodology, age ranges in samples or the specific forms of CBT employed across studies. In the present study, most of the CBT was delivered online and it is possible that the components targeting attention, such as exposure, was less effective than if they had been delivered face-to-face with a therapist, something that might have affected the results.” 

9. Why is vigilance toward threat predictive of greater symptom reduction (line 162-163)?

Reply: The suggested hypothesis is that those who are more vigilant to threat are also more likely to attend closely to threat during exposure training in CBT. And that attention on the feared object is important for habituation and anxiety symptom reductions. To emphasize this, the follow information has been added in the background (page 8):

“The same has been reported in a homogenous sample of adults with SAD (32), and a suggested explanation is that those who attend more to threat during exposure excercises will experience more habituation and thereby greater alleviation of symptoms”

10. The authors may consider rephrasing “whether it is more beneficial” (line 169) to summarizing the link between the described bias and positive intervention outcome.

Reply: We welcome this suggestion and have now changed the sentence to:

“In summary, it is not clear if an attention bias towards or away from threat, prior to entering therapy for SAD, will increase the likelihood of a positive treatment outcome.”

Method:

1. The authors state that all SAD participants fulfilled DSM-5 criteria for a principal diagnosis of SAD in Participants section. May be helpful for clarification to include the name of the interview and that it was used for both groups (so the reader does not have to search for this info in Materials section). Relatedly, why were two structured interviews used in the SAD group used?

Reply: We have now added information in the Participant section about the specific diagnostic screening tool used. There was only one interview used for the SAD group, but as the social anxiety section in MINI-KID is quite brief, we supplemented this section with the questions from the social anxiety section in the ADIS interview. In order to get a better basis for determining diagnostic status regarding SAD specifically.

2. The authors have designed the experimental testing of attention bias through use of a gap-overlap task. It is unclear if the gap and overlap trials were run together or in separate blocks.

Reply: Gap and overlap trials were presented together. We have now clarified this (page 15).

3. Additional citations are needed in the Experimental Paradigm section (e.g., lines 267-269; 271-275).

Reply: This is a good point, we agree and have now added the two following references regarding the gap-overlap paradigm and impaired disengagement/avoidance (page 14):

Van der Stigchel S, Hessels RS, van Elst JC, Kemner C. The disengagement of visual attention in the gap paradigm across adolescence. Experimental brain research. 2017; 235(12):3585-3592.

Moriya J, Tanno Y. The time course of attentional disengagement from angry faces in social anxiety. Journal of behavior therapy and experimental psychiatry. 2011;42(1):122-128.

Discussion

1. The Discussion section is opened with the “main purpose” of the study, which leaves out half of the research questions addressed.

Reply: Thank you for pointing this out, we have now changed the wording as there it not really any main or secondary purposes of the study (page 22):

”The main purpose of this study was to compare visual attention patterns in adolescents with SAD, with a non-anxious group, as well as to explore the associations between attention biases and CBT”

2. Are there any differences in emotion stimuli used between this study and others in the literature? Could that have an impact on findings of no group differences in both gap and overlap tasks?

Reply: This is a very interesting question, there are a few different stimuli sets used in the studies within the field. We used the Karolinska Directed Emotional Faces (KDEF) package, whereas the studies we refer to in the manuscript used pictures of faces from sources such as “MacArthur NimStim Face Stimulus Set” and the “Ekman pictures of facial affect”. It is difficult to determine how much this influenced the outcome. One key aspect is of course that the participants recognize the different emotions that the faces express on the pictures. In our study, we showed all pictures used in the gap- overlap trials (at the very end of the eyetracking experiment) and asked the participants to describe the emotion on the faces. Most participants were able to state the correct emotion for almost all of the faces so we feel fairly confident that unclear/vague stimuli were not a source of error in this trial. Furthermore, since the same actors posed for all the included emotions, we were able to control for potential effects of basic facial features such as distance between the eyes, or the shape of specific features on the between-conditions effects. But it is of course difficult to rule out that the choice of stimuli set did not have any sort of impact on the outcome. 

3. Is there some sort of developmental process in adolescence occurring that could account for lack of group differences between NA and SAD, especially in light of finding of differences during other developmental periods (childhood, adulthood)?

Reply: This question taps into question #3 (Introduction) by the same reviewer and we hope to have addressed this in our response to that comment. Briefly, the difference between anxious and non-anxious groups that has been shown to be quite distinct in adult populations, does not seem to be so evident in youth populations. Particularly with younger children, it seems as if there are no differences between anxious and non-anxious groups (with regard to vigilance to threat), wherefore it has been assumed that a developmental process occurring during later childhood/adolescence is responsible for the fact that the two groups gradually start to distinguish themselves from each other. 

4. The authors don’t provide information on co-occurring anxiety diagnoses and this would be helpful. Could this impact the interpretation of results? For example, could co-occurring diagnosis (anxiety or otherwise) impact why there are no findings with regard prediction of treatment outcome or to changes in attention processes following treatment? Please address in Method/Discussion.

Reply: This is a very good point and we have now added information about this in the participants section (page 10):

“In the SAD group, 14 participants (56%) had a comorbid disorder and the most common ones were specific phobia (7 participants, 28%) and generalized anxiety disorder (5 participants, 20%).”

As well as brought it up in the discussion (page 27):

“Lastly, as participants were recruited from a clinical trial, about half of the sample had a comorbid psychiatric disorder and although these were secondary to SAD, it may have influenced the results. It is, however, very common with comorbidity in the SAD population and a strictly homogenous SAD sample would also have had implications for the generalizability of the findings.”

5. What could the potential impact be of ICBT intervention, versus more traditional in-person CBT, be on findings? For example, is there less emphasis on exposure, and could this impact the findings of no change in attentional processes post-treatment?

Reply: This question was, we believe, responded to in the reply to the same reviewer’s comment #8 (Introduction). 

Minor points to address:

1. The sentence, “One of the advantages…” (lines 101-103) is not clear. Eye tracking does not measure shifts “over time” as trials are averaged. Please clarify what is meant here. The sentence, “This practice has been established…” (lines 120-121) is not clear. Please rephrase to clarify point.

Reply: We agree that the first sentence is unclear and we have now removed the last part of it (“…thus allowing a more detailed study of attention shifts over time”). 

With regard to the second sentence, we also agree that it was not clear and have now changed it (page 6):

”To use one form of presumably threatening stimuli for individuals with different anxiety disorders could be problematic as disorder-congruent threats are known to be associated with more attention bias than generally threatening stimuli (17).”

2. Please separate paragraph starting at line 118, starting with “ There are, however, few….

Reply: This has now been done.

3. There are several typos and manuscript needs to be proof read; for example, line 161 is missing youth “with” SAD.

Reply: Thank you for pointing this out, we have now read the manuscript carefully and corrected all typos we were able to find.

4. Please identify that the numbers reported in Table 2 are in milliseconds.

Reply: This has now been done.

5. Figures 3 & 4: Please consider adding indicators of significant results where they exist.

Reply: This has now been done.

---

## [Decision Letter · Decision Letter 1]

10 Oct 2019

PONE-D-19-18168R1

Visual attention to emotional faces in adolescents with social anxiety disorder receiving cognitive behavioral therapy

PLOS ONE

Dear Dr Högström,

Thank you for submitting your manuscript to PLOS ONE. After careful consideration, we feel that it has merit but does not fully meet PLOS ONE’s publication criteria as it currently stands. Therefore, we invite you to submit a revised version of the manuscript that addresses the points raised during the review process.

You will see that there are comments from a new reviewer, a statistician who was asked directly by PLoS to review your paper re: its statistical aspects. Please address these comments carefully, in addition to those still raised by one of the previous reviewers.

We would appreciate receiving your revised manuscript by Nov 24 2019 11:59PM. To enhance the reproducibility of your results, we recommend that if applicable you deposit your laboratory protocols in protocols.io, where a protocol can be assigned its own identifier (DOI) such that it can be cited independently in the future. For instructions see: http://journals.plos.org/plosone/s/submission-guidelines#loc-laboratory-protocols

We look forward to receiving your revised manuscript.

Kind regards,

Nouchine Hadjikhani, MD, PhD

Academic Editor

PLOS ONE

Reviewers' comments:

Reviewer's Responses to Questions

**Comments to the Author**

1. If the authors have adequately addressed your comments raised in a previous round of review and you feel that this manuscript is now acceptable for publication, you may indicate that here to bypass the “Comments to the Author” section, enter your conflict of interest statement in the “Confidential to Editor” section, and submit your "Accept" recommendation.

Reviewer #1: All comments have been addressed

Reviewer #2: (No Response)

Reviewer #3: (No Response)

2. Is the manuscript technically sound, and do the data support the conclusions?

Reviewer #1: Yes

Reviewer #2: Yes

Reviewer #3: Partly

3. Has the statistical analysis been performed appropriately and rigorously? 

Reviewer #1: Yes

Reviewer #2: Yes

Reviewer #3: No

4. Have the authors made all data underlying the findings in their manuscript fully available?

Reviewer #1: Yes

Reviewer #2: Yes

Reviewer #3: Yes

5. Is the manuscript presented in an intelligible fashion and written in standard English?

Reviewer #1: Yes

Reviewer #2: Yes

Reviewer #3: Yes

6. Review Comments to the Author

Reviewer #1: (No Response)

Reviewer #2: Overall, the authors provided very helpful responses to the reviewer comments and the manuscript has benefitted substantially. Thank you for the thoughtful consideration of the points raised. Several additional/follow-up suggestions for revision follow:

While the authors now define vigilance and avoidance in the Introduction, it is still somewhat confusing in the Abstract (especially with regard to vigilance). It is suggested that any definitions across Abstract and Introduction be consistent.

The authors may consider changing use of “normally developed children” to “typically developing” or “non-anxious children” throughout the manuscript.

Line 126: “looks to maintain” is not clear.

While the definitions of vigilance and avoidance (lines 127-129) are helpful, additional clarity would be of benefit. It appears that vigilance is measured by orienting (toward threat – is this measured by Gap trials) and avoidance measured by disengagement (away from threat – is this measured by overlap trials). Perhaps these concepts could still be more clearly/concisely delineated throughout the manuscript. For example, no mention of vigilance in results (lines 400-409) – is this what was measured here?

Lines 318-319: The use of “impaired disengagement” twice is redundant.

Lines 485-487: Are these robust traits or are these not effectively targeted in the ICBT intervention (as outlined in lines 554-557)?

Reviewer #3: The manuscript entitled 'Visual attention to emotional faces in adolescents with social anxiety disorder receiving cognitive behavioral therapy' with the aim to investigate the attention bias in youth with SAD and the association with outcome from CBT.

This is quite an interesting study. The manuscript can be further improved based on the following comments.

Methods

Line 241, more information to be provided on 'population register'.

Other information apart from Cohen'd, power 80% that was used to derive the calculated sample size to be stated e.g. group involved, alpha etc.

Table 1, in the footnote, the specific t test to be named. the word 'test' to be added for chi-square.

The name of the statistical software including the version and publisher name that was used for the data analysis to be stated.

The acceptance level of significance to be stated.

Results

The word significant to be added to the word difference(s) and other related interpretation in the results and discussion section where applicable.

Table 1 and Figure 1,2,3,4 size to be enlarged as it is difficult for the reader to visualize.

For all F statistics presentation, the dfs between them to be separated with a space after comma.

Line 411, the post hoc analyses refers to, with or without correction? This to be stated in the statistical analyses section.

Line 392 - 448, all these results could be displayed in one table form but with different section according to the domains for easy visualization.

Symbol =< to be replaced with ≤

Table 3, BF01 and BF10 to be clearly defined in the footnote. It would be good to illustrate for the reader one example how the BF value was obtained including BF formula and changes (Δ) in BIC.

Figure 3 and 4, error bar and n to be stated.

Some references did not conform to the journal format.

7. PLOS authors have the option to publish the peer review history of their article (what does this mean?). If published, this will include your full peer review and any attached files.

Reviewer #1: No

Reviewer #2: No

Reviewer #3: No

---

## [Author Response · Author response to Decision Letter 1]

6 Nov 2019

Response to Reviewers

Dear Dr Hadjikhani,

Thank you for allowing us to revise and resubmit the manuscript “Visual attention to emotional faces in adolescents with social anxiety disorder receiving cognitive behavioral therapy” (PONE-D-19-18168R1). We have again done our best to address the very constructive and helpful feedback received from the reviewers. 

Best regards

Jens Högström

PhD, Lic. Psychologist 

Karolinska Institutet

Child and Adolescent Psychiatry

Clinical Research Unit

Gävlegatan 22 | 113 30 Stockholm

E-mail: jens.hogstrom@ki.se

Phone: 0046 8 514 522 08

Review Comments to the Author

Reviewer #2: Overall, the authors provided very helpful responses to the reviewer comments and the manuscript has benefitted substantially. Thank you for the thoughtful consideration of the points raised. Several additional/follow-up suggestions for revision follow:

Reply: Thank you very much and we appreciate the effort that the reviewer is putting into helping us improve the manuscript.

1. While the authors now define vigilance and avoidance in the Introduction, it is still somewhat confusing in the Abstract (especially with regard to vigilance). It is suggested that any definitions across Abstract and Introduction be consistent.

Reply: We agree with the reviewer that a further streamlining between the abstract and introduction was warranted regarding this. We have now changed the following part of the abstract (page 3, new text in italics):

”Latency to attend to pictures of faces with different emotions (vigilance) and latency to disengage from social stimuli (avoidance) was examined in N=25 adolescents (aged 13-17) with SAD in relation to treatment outcome.”

Along with the subsequent operationalization in the abstract, we hope that it is now clear to the reader what we mean by vigilance and avoidance in this study:

”Vigilance was operationalized as the time it took to relocate the gaze from a central position to a peripherally appearing social stimulus. The latency to disengage from a centrally located social stimulus, when a non-social stimulus appeared in the periphery, was used as a proxy for avoidance.”

2. The authors may consider changing use of “normally developed children” to “typically developing” or “non-anxious children” throughout the manuscript.

Reply: This is a welcome suggestion and we have now made changes in the manuscript accordingly.

3. Line 126: “looks to maintain” is not clear.

Reply: We agree and the sentence has now been changed into (page 6):

“With maturation though, typically developed children learn to inhibit automatic responding and gradually become less reactive to threatening stimuli, whereas those with an anxiety disorder are more likely to maintain the biased attention towards threat (13).”

4. While the definitions of vigilance and avoidance (lines 127-129) are helpful, additional clarity would be of benefit. It appears that vigilance is measured by orienting (toward threat – is this measured by Gap trials) and avoidance measured by disengagement (away from threat – is this measured by overlap trials). Perhaps these concepts could still be more clearly/concisely delineated throughout the manuscript. For example, no mention of vigilance in results (lines 400-409) – is this what was measured here?

Reply: It is correctly understood that vigilance is measured by orienting to threat, in gap trials, and that avoidance is measured by disengagement from threat, in overlap trials. To be consistent, we agree with the reviewer that this should be delineated throughout the manuscript. We have now changed the headlines in the result section where we report analyses of gap- and overlap trials (page 19):

 “Latency to attend to peripheral emotional stimuli (vigilance - measured with gap trials)” 

And: 

”Disengagement from central emotional stimuli (avoidance – measured with overlap trials)”

We have furthermore clarified and added information in the following parts of the result section (pages 19-21):

”Both groups thus attended faster to peripherally appearing angry faces compared to happy and neutral faces, and thereby showed vigilance to threat.”

And:

”In summary, both groups were thus faster to disengage from angry faces compared to happy and neutral faces, and thereby showed avoidance of threat”

And:

“With regard to vigilance and avoidance, analyses showed that there was no main effect of time on latency to attend to peripheral stimuli (F (2,35.84)=0.38, p=.68) or on latency to disengage from central stimuli (F(2,35.77)=0.13, p=.88).”

As well as in the following part of the discussion (pages 22-23):

”Results demonstrated that both the SAD and the NA groups were more vigilant towards angry faces compared to neutral and happy faces, indicated by a tendency for both groups to attend faster to appearing social threats”

We hope that these changes and additional information makes the manuscript more consistent with regards to how vigilance and avoidance was operationalized in the study.

5. Lines 318-319: The use of “impaired disengagement” twice is redundant.

Reply: Thank you for noticing this mistake, it has now ben corrected.

6. Lines 485-487: Are these robust traits or are these not effectively targeted in the ICBT intervention (as outlined in lines 554-557)?

´

Reply: This is a good point, and we admit that we cant be certain to what extent attention biases in this group are stable traits based on the findings from this study. We have therefore softened the conclusion in the following statement, in the discussion (page 23):

“Lastly, vigilance and disengagement latency in the SAD group did not change over the course of treatment indicating that visual attention biases in SAD are could be fairly robust traits.”

We believe that this way of expressing ourselves is more in line with the discussion that follows later (page 26) where the possible impications of different forms of CBT (on impact on attention), are addressed.

Reviewer #3: The manuscript entitled 'Visual attention to emotional faces in adolescents with social anxiety disorder receiving cognitive behavioral therapy' with the aim to investigate the attention bias in youth with SAD and the association with outcome from CBT.

This is quite an interesting study. The manuscript can be further improved based on the following comments.

Reply: Thank you!

Methods

1. Line 241, more information to be provided on 'population register'.

Reply: This is a welcome suggestion and the following information has now been added (page 10):

“Participants in the non-anxious (NA) group (N=22, age 13-18) were recruited from 450 randomly selected adolescents in the population register. This register is administered by the Swedish Tax Agency and provides services to government institutions, including universities, that are in need of population data such as personal identification numbers and addresses to groups of individuals. Participants in the NA group were matched with the SAD participants on age and gender.”

2. Other information apart from Cohen'd, power 80% that was used to derive the calculated sample size to be stated e.g. group involved, alpha etc.

Reply: Thank you for pointing this out, we have now added the following information to the description of the power analysis (page 11):

“The sample size was similar to, or slightly larger than, most previous eye-tracking studies investigating social attention (11), with 80% power, given alpha 0.05, two-tailed, to detect medium effect sizes (corresponding to Cohen’s d = .7 for comparisons between the SAD and NA group and d = .6 for within-group analyses in the SAD group).”

3. Table 1, in the footnote, the specific t test to be named. the word 'test' to be added for chi-square.

Reply: We have now added this requested information to the footnote in table 1 (page 12):

”Independent means t-test / chi-square test comparisons between groups”

4. The name of the statistical software including the version and publisher name that was used for the data analysis to be stated.

Reply: We have now added the following information to the statistical analyses paragraph (page 17):

”Statistical analyses were conducted in MATLAB version R2018b (Mathworks, Inc).”

5. The acceptance level of significance to be stated.

Reply: We have now added the following line to the statistical analyses paragraph (page 16):

“An alpha level of p<.05 indicated statistical significance.”

Results

6. The word significant to be added to the word difference(s) and other related interpretation in the results and discussion section where applicable.

Reply: The word “significant” has now been added at locations where we describe significant differences, throughout the result and discussion section.

7. Table 1 and Figure 1,2,3,4 size to be enlarged as it is difficult for the reader to visualize.

Reply: Table 1, 2 and all figures have now been enlarged for better visibility. If they need to be adjusted further we are also happy to supply them in any format of your convenience so that they can be edited optimally. 

8. For all F statistics presentation, the dfs between them to be separated with a space after comma.

Reply: This has now been changed accordingly.

9. Line 411, the post hoc analyses refers to, with or without correction? This to be stated in the statistical analyses section.

Reply: P-values for follow-up tests are reported without correction. This is now stated in the statistical analyses section (page .

10. Line 392 - 448, all these results could be displayed in one table form but with different section according to the domains for easy visualization.

Reply: We agree that these results could be displayed in a table and we have given it some consideration. However, as we already have three tables and four figures in the manuscript, we feel that it might be sufficient as it is. And with the new clarifications in this part of the result section, we do believe that the reader will be able to grasp and digest the information in a reasonably good way. However, if the reviewer and/or the editor has a strong opinion about this, we will of course abide to the recommendation. 

11. Symbol =< to be replaced with ≤

Reply: This has now been changed accordingly.

12. Table 3, BF01 and BF10 to be clearly defined in the footnote. It would be good to illustrate for the reader one example how the BF value was obtained including BF formula and changes (Δ) in BIC.

Reply: Thank you for this suggestion. We have now added this information in a footnote (page 17).

13. Figure 3 and 4, error bar and n to be stated.

Reply: This information has now been added to the legend in figure 3 and 4.

14. Some references did not conform to the journal format.

Reply: We have now gone through all references thoroughly ta make sure that they conform to the journal format.

---

## [Editor Report · Decision Letter 2]

8 Nov 2019

Visual attention to emotional faces in adolescents with social anxiety disorder receiving cognitive behavioral therapy

PONE-D-19-18168R2

Dear Dr. Högström,

We are pleased to inform you that your manuscript has been judged scientifically suitable for publication and will be formally accepted for publication once it complies with all outstanding technical requirements.

With kind regards,

Nouchine Hadjikhani, MD, PhD

Academic Editor

PLOS ONE
---

## [Editor Report · Acceptance letter]

15 Nov 2019

PONE-D-19-18168R2 

Visual attention to emotional faces in adolescents with social anxiety disorder receiving cognitive behavioral therapy 

Dear Dr. Högström:

I am pleased to inform you that your manuscript has been deemed suitable for publication in PLOS ONE. Congratulations! Your manuscript is now with our production department. 

With kind regards,

on behalf of

Prof. Nouchine Hadjikhani 

Academic Editor

PLOS ONE